# Inflammation in Penile Squamous Cell Carcinoma: A Comprehensive Review

**DOI:** 10.3390/ijms26062785

**Published:** 2025-03-19

**Authors:** Mateusz Czajkowski, Piotr M. Wierzbicki, Maciej Dolny, Marcin Matuszewski, Oliver W. Hakenberg

**Affiliations:** 1Department of Urology, Medical University of Gdańsk, Mariana Smoluchowskiego 17 Street, 80-214 Gdansk, Poland; maciej.dolny95@gmail.com (M.D.); matmar@gumed.edu.pl (M.M.); 2Department of Histology, Medical University of Gdańsk, Dębinki, 80-211 Gdansk, Poland; pwierzb@gumed.edu.pl; 3Department of Urology, University Medical Center Rostock, 18055 Rostock, Germany; oliver.hakenberg@med.uni-rostock.de; 4Department of Urology, Jena University Hospital, 07747 Jena, Germany

**Keywords:** penile cancer, inflammation, tumor immune microenvironment, cytokines, HPV

## Abstract

Inflammation appears to play a crucial role in the development and progression of penile cancer (PeCa). Two molecular pathways of PeCa are currently described: HPV-dependent and HPV-independent. The tumor immune microenvironment (TIME) of PeCa is characterized by the presence of tumor-associated macrophages, cancer-associated fibroblasts, and tumor-infiltrating lymphocytes. The components of the TIME produce pro-inflammatory cytokines and chemokines, which have been found to be overexpressed in PeCa tissues and are associated with tumor progression and unfavorable prognoses. Additionally, the nuclear factor kappa B (NF-κB) pathway and secreted phosphoprotein 1 (SPP1) have been implicated in PeCa pathogenesis. Elevated C-reactive protein (CRP) levels and the neutrophil-to-lymphocyte ratio (NLR) have been identified as potential prognostic biomarkers in PeCa. This overview presents the complex contribution of the inflammatory process and collates projects aimed at modulating TIME in PeCa.

## 1. Introduction

Penile cancer (PeCa) is rare in Western European nations, with a constant incidence rate of approximately 0.8 cases per 100,000 male inhabitants over the past three decades. However, this malignancy exhibits higher prevalence in African and South American countries, where it accounts for up to 10% of all cancer diagnoses. Despite its classification as a rare disease in developed countries, PeCa has a high mortality rate. In 2020, the global incidence and mortality rates of PeCa were 36.068 and 13.211, respectively. In terms of histopathological structure, more than 95% of PeCa cases are classified as squamous cell carcinoma [1,2,3]. Early detection remains crucial, as diagnosing PeCa in stages I and II results in a five-year survival rate of up to 85% following surgical intervention. Conversely, when identified in stages III and IV, the survival rate decreases to 59%; it further diminishes to 11% when distant metastases are present [4]. In stage IV of PeCa, surgical interventions in combination with neoadjuvant or adjuvant platinum-based chemotherapy demonstrate insufficient efficacy, with 63.3% of patients experiencing tumor recurrence or progression with a median survival of only 5.6 months [5]. Furthermore, an effective second-line chemotherapy regimen has not been established [6,7]. Thus, it is essential to understand the pathways that contribute to the development and progression of this disease, particularly the relationship between inflammatory reactions, to enable the potential use of anti-inflammatory treatments. The identification of novel biomarkers could potentially enhance early detection strategies and facilitate more personalized treatment approaches, potentially contributing to improved survival rates across all stages of PeCa.

Inflammation is a natural defense mechanism that occurs in response to harmful stimuli, including chemical, physical, and biological agents. This process stimulates cell growth and activates the host immune system, thereby leading to cell damage. However, chronic inflammation can lead to various diseases, including cancer [8,9,10]. PeCa is induced by chronic inflammatory processes in about 50% of cases. The idea that inflammation can play a role in cancer development was first proposed by Virchow in 1863. According to Virchow’s hypothesis, chronic inflammation and leukocyte infiltration are the initial steps in this process [10]. The TIME encompasses several processes, resulting in the presence of tumor-associated macrophages (TAMs), cancer-associated fibroblasts (CAFs), endothelial cells, tumor-infiltrating lymphocytes (TILs), and cytokines, which collectively contribute to the dynamic interplay between the tumor and the immune system [6,11].

Numerous studies have demonstrated that inflammation contributes to the development of various types of cancer, such as those affecting the urogenital system [12]. One of the most well-documented examples is bladder cancer, in which chronic inflammation resulting from urogenital schistosomiasis is a recognized risk factor [13]. Renal cell carcinoma (RCC) is another type of urogenital cancer, in which inflammatory factors play a significant role. Furthermore, an increasing number of studies have confirmed that high production levels of many pro-inflammatory molecules are associated with a poor prognosis [14,15]. The primary risk factors for PeCa include phimosis, lichen sclerosus, chronic inflammation of the glans and foreskin, a history of smoking, and human papillomavirus (HPV) infection [1]. Research has documented increased levels of pro-inflammatory cytokines, such as interleukins IL-1A, IL-6, and INF-Υ, in individuals with phimosis and lichen sclerosus [16]. Additionally, a recent meta-analysis demonstrated that 50.8% of PeCa cases are in HPV-positive patients, which is associated with a local immune response against the virus [17]. Furthermore, cells containing HPV genetic material demonstrate elevated nitric oxide (NO) levels, which promote inflammation and DNA damage [18]. Collectively, these findings suggest that inflammation plays a significant role in the etiology of PeCa.

This review aims to shed light on the inflammatory mechanisms that are crucial in the development and progression of PeCa. Furthermore, we examine the prognostic value of inflammatory biomarkers of this disease.

## 2. Tumor-Infiltrating Lymphocytes, Macrophages, and Fibroblasts: Key Players in the Cancer Immune Microenvironment

The tumor immune microenvironment (TIME) is the cellular microenvironment that surrounds cancer stem cells, as shown in Figure 1. The TIME is widely recognized as a critical factor in tumor formation due to its role in harboring cancer cells that communicate with neighboring cells via the circulatory and lymphatic systems, thereby reacting to and influencing cancer development and progression. Furthermore, non-cancerous cells within this microenvironment play essential roles throughout all stages of cancer development by promoting and supporting unregulated cell division [19,20]. The TIME comprises various cellular components, including endothelial cells, immune cells, and fibroblasts. Endothelial cells are essential for tumor growth and protect tumor cells against host immune responses. These cells typically form new blood vessels, thereby supplying essential nutrients for tumor expansion [20,21]. The second major component consists of immune cells, including macrophages, granulocytes, and lymphocytes. These cells participate in diverse immune functions and activities, such as sustaining the inflammatory processes induced by tumors that promote cancer survival [22]. Macrophages, the most prevalent immune cell type in the TIME, play multiple roles in cancer progression. They facilitate tumor cell extravasation into the bloodstream, suppress antitumor immune mechanisms, and aid circulating cancer cells in colonizing distant lymph nodes and organs, thus leading to metastatic growth. Tumor-associated macrophages (TAMs) enhance, facilitate, or counteract the antitumor effects of radiation, cytotoxic drugs, and immune checkpoint inhibitors (ICIs) [23]. The final component of the TIME is fibroblasts, which enable cancer cells to migrate from the primary tumor site to the circulatory system and facilitate systemic metastasis [21].

### 2.1. Tumor-Infiltrating Lymphocytes

Tumor-infiltrating lymphocytes (TILs), a heterogeneous group primarily comprising T, B, and NK cells, play a crucial role in modulating the immune response associated with cancer. The composition of this cell mixture significantly influences the progression or suppression of tumor cells, and thus, the prognosis and treatment outcomes in various cancers. The mechanism of action is typically complex, and the relationships among the various types of lymphocytes are multifaceted. Cytotoxic CD8^+^ T cells play a crucial role in the immune response against tumor cells. These specialized cells recognize specific antigens present on the surfaces of tumor cells, triggering a complex immune cascade. The process follows a characteristic pathway, involving the initial recognition, activation, and subsequent elimination of target cells [24]. CD8^+^ T cells in the TIME are typically supported by CD4^+^ T helper 1 (Th1) cells, which secrete important cytokines such as interferon-gamma (IFN-γ) and interleukin-2 (IL-2) (Figure 1). These cytokines enhance the activation and proliferation of CD8^+^ T cells, further amplifying the antitumor immune response [19,25]. While CD8^+^ T cells and Th1 cells are primarily associated with antitumor immunity, other CD4^+^ T-cell subsets (Th2) contribute to the overall immune landscape in diverse ways. Th2 cells, for instance, support B-cell response through the production of cytokines such as IL-4 and IL-5, which are involved in humoral immunity [24]. Th17 cells, through the secretion of distinct cytokines such as IL-17, IL-21, and IL-22, exhibit contradictory effects on carcinogenesis. In colorectal, pancreatic, and non-small-cell lung cancers, Th17-related factors are associated with adverse outcomes. Conversely, in breast cancer, IL-22 production is correlated with reduced tumor formation and improved prognosis. Additionally, in ovarian cancer, elevated levels of tumor-associated IL-17 are predictive of improved patient survival [26]. This phenomenon illustrates the complex and occasionally conflicting roles that different T-cell subsets can exhibit in the context of tumor immunology, emphasizing the need for a nuanced understanding of immune cell interactions within the TIME. The B lymphocytes in the TIME play complex and sometimes opposing roles in cancer progression. These cells can influence tumor cell survival, proliferation, and treatment resistance, while also potentially facilitating immune escape mechanisms. The precise impact of B cells on cancer development and tumor suppression remains a subject of debate within the scientific community. B-cell infiltration is associated with a favorable prognosis in breast and ovarian cancers. Furthermore, research has demonstrated that modulating B-cell activity within the tumor immune microenvironment (TIME) can disrupt cancer-induced immunosuppressive events. Conversely, genetic mouse models of skin cancer present contrasting results, indicating a tumor-promoting role for B cells. One potential mechanism may involve the TGF-β-dependent conversion of Forkhead box P3 (FOXP3) cells, which can support and promote metastasis [27,28]. In contrast, natural killer (NK) cells demonstrate antitumor functions. These immune cells are equipped with a variety of receptors that enable them to recognize and eliminate tumor cells while sparing healthy tissues. NK cells serve as a crucial component of the body’s initial defense against cancer, utilizing their cytotoxic capabilities to directly eliminate malignant cells, mainly by secreting TNF-α, Granzyme A, Granzyme B, and perforin after the recognition of antibodies produced by B cells of plasma cells [29] (Figure 1).

Pereira et al. demonstrated that elevated CD3^+^ TIL levels were significantly associated with improved survival outcomes and could potentially serve as prognostic indicators in gastric adenocarcinoma [30]. Furthermore, a recent review by Brummel et al. stated that both CD4^+^ and CD8^+^103^+^ TILs populations are associated with increased overall survival (OS) in ovarian and cervical cancers [31]. In contrast, in a study of 50 patients with RCC, Sanders et al. found an association between high numbers of CD103^+^ TILs and poor survival rates [32]. Similarly, in an investigation of TILs in cutaneous SCC, Lai et al. found that a high proportion of CD8^+^103^+^ TILs was associated with poor outcomes [33].

TILs also play an important role in the TIME of PeCas and are associated with their prognosis. It has been observed that a high level of inflammation and a decrease in the number of FOXP3^+^ T cells, also known as Tregs, which are responsible for regulating other lymphocytes and suppressing antitumor responses, are linked to favorable outcomes in PeCa [6,34,35]. In contrast, higher levels of infiltration of CD8^+^ TILs in the tumor-associated stroma are associated with lymph node metastasis in PeCa [11]. CD20^+^ B-cell lymphocytes, which are predominantly involved in humoral immunity, appear to be of relatively low significance in the TIME of PeCa compared with CD8^+^ or FOXP3^+^ lymphocytes. According to Vassallo et al., only 16.4% of PeCa samples exhibit moderate-to-high densities of CD20^+^ lymphocytes [35]. However, a study conducted by Hladek et al. showed increased densities of CD3^+^, CD8^+^, and CD20^+^ lymphocytes in PeCa tissues, highlighting the complexity of the PeCa TIME [36].

One of the most significant regulatory mechanisms influencing the activity of TILs is the programmed cell death protein 1/programmed cell death ligand 1 (PD-1/PD-L1) axis [37]. PD-1 is a cell surface receptor expressed on various immune cells, including CD8+ T cells, CD4+ T cells, NK cells, monocytes, antigen-presenting cells (APCs), and CD20^+^ lymphocytes [38]. It plays a crucial role in regulating TIL activity by interacting with its cognate ligand, PD-L1. PD-L1, a membrane-bound protein, forms a complex with PD-1 mainly on CD8^+^ T cells, leading to the inhibition of immune responses. This mechanism is also employed by cancers as an immune evasion strategy to inhibit the host’s antitumor immune response [39]. Consequently, the PD-1/PD-L1 axis is a primary target of immunotherapy in various cancers, including PeCa. The mechanism of immune checkpoint inhibitors (ICIs) generally involves blocking the binding of cancer’s PD-L1, thereby facilitating the patient’s immune response against cancer [40].

The ongoing clinical trials and studies investigating ICIs for PeCa treatment have shown mixed results, with some promising outcomes but also limitations. The PERICLES Study, a Phase II trial of atezolizumab (anti-PD-L1) with or without radiotherapy, demonstrated a one-year progression-free survival (PFS) rate of 12.5% among 32 participants, with a median overall survival (OS) of 11.3 months and an overall response rate (ORR) of 16.7% [41]. Similarly, the ORPHEUS Study, a Phase 2 trial of retifanlimab (anti-PD-1), reported an ORR of 16.7% in 18 patients, with a median PFS of 2.0 months and a median OS of 7.2 months [42]. Other studies have explored combination therapies and different ICIs. A basket trial by Apolo et al. investigated the combination of cabozantinib, nivolumab, and ipilimumab, showing an ORR of 44.4% in nine penile cancer patients, with a median PFS and OS of 4.8 and 6.7 months, respectively [43]. A retrospective analysis by Rouvinov et al. reported near-complete response in three patients treated with cemiplimab (anti-PD-1) [44]. Ongoing trials, such as the EPIC trial studying cemiplimab, continue to explore the potential of ICIs in penile cancer treatment [45]. However, some basket trials, including those by McGregor et al. and Naing et al., have shown limited or no response in penile cancer patients [46,47]. These studies highlight the need for further research with larger sample sizes to better understand the efficacy of ICI therapy in penile cancer and identify potential biomarkers for patient selection. Therefore, the primary clinical implication of TILs in PeCa lies in their potential use as a prognostic biomarker. Furthermore, TILs could serve as targets for immunotherapy, either alone or in conjunction with traditional chemotherapy. Moreover, the evaluation of the concentration of specific types of TILs in PeCa may serve as a predictive biomarker for the response to immunotherapy. This assessment facilitates the identification of patients who are more likely to benefit from such treatments. Additionally, understanding the composition and distribution of TILs within the TIME could provide insights into the mechanisms of immune evasion employed by PeCa cells, potentially leading to the development of novel therapeutic approaches. Moreover, the longitudinal monitoring of TIL levels during treatment in clinical trials could offer valuable information about the dynamic immune response, which may change PeCa patients’ prognosis.

### 2.2. Tumor-Associated Macrophages

The role of macrophages in the TIME is also multifaceted and complex. Tumor-associated macrophages (TAMs) primarily originate from peripheral inflammatory monocytes (Figure 1). These monocytes are recruited to the tumor site through a complex interplay of the cytokines and chemokines secreted by tumor cells. Key mediators in this process include colony-stimulating factor 1 (CSF-1) and members of the vascular endothelial growth factor (VEGF) family, which function as cytokines, as well as chemokines such as C–C motif chemokine 2 (CCL) 2 and CCL5 (Figure 1). Upon stimulation, these monocytes migrate through the bloodstream and infiltrate the tumor tissues, where they undergo further differentiation to become TAMs. While CSF-1 recruits monocytes to tumors, promotes macrophage survival, and induces TAMs to suppress the immune system, granulocyte–macrophage colony-stimulating factor (GM-CSF) exhibits a contrasting effect by stimulating macrophage proliferation and activating antitumor functions [48,49] (Figure 1). Previously, it was hypothesized that their primary function was to contribute to antitumor immunity alone. However, clinical evidence suggests that TAMs can promote tumor growth and malignancy [50]. TAMs are categorized into two populations: M1 and M2 (Figure 1). Interferon-gamma (IFN-γ) plays a crucial role in the polarization of tumor-associated macrophages (TAMs) towards an M1-like phenotype in the TIME. This polarization is characterized by the induction of a Th1-type immune response, which is essential for the innate host defense and antitumor immunity. M1 TAMs exhibit enhanced antigen presentation capabilities through the increased expression of major histocompatibility complex class I and II molecules (MHC-I and MHC-II), as well as the co-stimulatory molecules CD80 and CD86. These macrophages also produce pro-inflammatory cytokines, including tumor necrosis factor-alpha (TNF-α), and they release inflammatory mediators such as reactive oxygen species (ROS) and nitric oxide, contributing to their antitumor functions [19,49,51] (Figure 1). In contrast, TAMs of the M2 phenotype are polarized in response to cytokines associated with Th2-mediated immune responses, such as IL-4 and IL-10. This polarization significantly impacts various physiological processes, including angiogenesis, anti-inflammatory responses, and immune regulation. M2 macrophages are identified by specific markers, such as the mannose receptor (CD206) and scavenger receptor with lower levels of MHC-II. These cells produce a diverse array of molecules, including arginase-1, matrix metalloproteinase-9 (MMP-9), VEGF, prostaglandin E2 (PGE2), transforming growth factor-beta (TGF-β), and anti-inflammatory cytokines such as IL-10 and IL-13 (Figure 1). The production of these factors contributes to tumor development through multiple mechanisms, highlighting the complex role of TAMs in the tumor microenvironment and their potential as targets for cancer therapy [52,53]. Both types participate in creating an environment that supports cancer invasion and spread. Nevertheless, M1 TAMs generally act as antitumorigenic agents, in contrast to M2 TAMs, which suppress the immune system and thus encourage cancer development [19]. Elevated TAM levels are associated with adverse outcomes in various cancers. Allison et al. reported that an elevated concentration of TAMs was associated with diminished OS or PFS in breast cancer. Moreover, their findings indicated that TAMs expressing CD163^+^ were more likely to predict survival outcomes than those expressing CD68^+^ [54]. Similarly, Lin et al. observed that elevated TAM levels were correlated with lymph node metastasis and FIGO stages in cervical cancer. Subsequently, they suggested that a high concentration of TAMs is indicative of poor prognosis in cervical cancer [55]. In squamous cell carcinomas of the head and neck, elevated levels of CD68^+^CD163^+^ TAMs are associated with decreased OS. Moreover, in these cancers, CD163^+^ serves as a robust prognostic indicator and is correlated with disease-free survival (DFS) and progression-free survival (PFS) [56]. In PeCa, TAMs are a significant component of the TIME and promote tumor growth. It is important to note that most tumor-associated macrophages in the PeCa TIME are CD68^+^CD163^+^ M2 macrophages. Interestingly, a high number of intramural CD163^+^ M2 macrophages is significantly associated with a higher incidence of lymph node metastasis (OR 2.45, *p* < 0.01) [11]. This could be attributed to the suppression of the TIL response and the promotion of tumor angiogenesis by CD163^+^ M2 macrophages [50,57]. In contrast, according to Chu et al., a significant number of CD68^+^ and CD206^+^ TAMs are associated with improved prognosis [6]. This may be because CD206^+^ is commonly regarded as a marker for M2 macrophages, although low CD206^+^ expression can also be observed in dendritic cells and specific types of lymphatic or endothelial cells. These conflicting data on the prognostic implications of high levels of M2 macrophages could be attributed to increased immune infiltration in the PeCa TIME [6]. Preclinical studies have suggested that CD206^+^ macrophages could potentially serve as targets for antitumor therapy [6,58].

### 2.3. Cancer-Associated Fibroblasts

Fibroblasts primarily originate from primitive mesenchyme, with some derived from neural crests [59]. Fibroblasts are major producers of extracellular matrix (ECM) and play crucial roles in tissue repair and wound healing. They can influence epithelial stem cell behavior, promote angiogenesis, and coordinate immune system functions. Fibroblastic reticular cells (FRCs) in lymph nodes create ECM conduits for antigen transit and leukocyte migration. Stellate cells, a specific type of fibroblast found in the liver and pancreas, are involved in metabolic homeostasis. Moreover, fibroblasts communicate with various cell types, performing diverse functions beyond ECM production [60]. The TIME contains various mechanisms that trigger the activation of cancer-associated fibroblasts (CAFs). Among these, the most important pathways are initiated by TGF-β, reactive oxygen species (ROS), pro-inflammatory cytokines (such as IL-1 and IL-6), and DNA damage [61] (Figure 1).

The difficulty of defining and characterizing CAFs highlights the challenges faced in understanding their role in tumor progression. While certain criteria have been established to identify CAFs, such as their elongated morphology, the absence of epithelial, endothelial, and leukocyte markers, and a lack of cancer-specific mutations, these parameters may not be sufficient to fully capture the heterogeneity and dynamic nature of these cells. The distinction between CAFs and other mesenchymal cell types, as well as cancer cells that have undergone epithelial-to-mesenchymal transition (EMT), further complicates their identification. The epithelial-to-mesenchymal transition (EMT) is a cellular program that transforms epithelial cells into mesenchymal-like cells. Notably, key differences between epithelial and mesenchymal cells include the presence of cell junctions, motility, and invasiveness. Therefore, EMT is associated with increased cell motility and invasiveness, which could contribute to cell separation from the primary tumor and facilitate metastasis. The primary functions of CAFs include depositing and remodeling the ECM, supporting metastasis by producing supportive matrix components, promoting tumor growth through growth factor and cytokine production, exerting immunosuppressive effects via cytokine production and antigen cross-presentation, and contributing to blood vessel formation through VEGF expression [60,62]. The study conducted by Goulet et al. provides significant insights into the role of CAFs and IL-6 in bladder cancer progression. Their findings demonstrate that CAFs are a primary source of IL-6 in the TIME, while bladder cancer cells express the IL-6 receptor (IL-6R). This paracrine signaling axis—which is a form of cellular communication in which a cell emits a signal to induce changes in nearby cells, thereby influencing their behavior—appears to play a significant role in promoting tumor growth and metastasis. The researchers observed that exposing bladder cancer cells to a CAF-conditioned medium significantly enhanced their proliferation, migration, and invasive capabilities, underscoring the importance of CAF-derived factors in cancer progression. Additionally, the study revealed a correlation between elevated IL-6 expression and more aggressive forms of bladder cancer, as well as an association with increased CAF presence in the tumor microenvironment [63]. Lopez et al. reported that in clear cell renal cell carcinoma (ccRCC), fibroblast activation protein (FAP), which has collagenolytic activity and is highly expressed on CAFs, was correlated with increased tumor size, higher grade, and a more advanced stage. Moreover, FAP expression was correlated with decreased survival rates [64]. Similarly, Errate et al. demonstrated a significant correlation between FAP immunostaining in the primary tumors of clear cell RCC and several adverse prognostic factors, including advanced tumor stage, high grade, and the presence of necrosis. Notably, FAP expression was also associated with decreased overall survival in ccRCC patients. These findings were consistent across various patient subgroups, including those with metastases at diagnosis and those who developed metastases during follow-up, underscoring the potential significance of FAP as a prognostic marker in ccRCC [65].

In the literature, there are scarce data regarding the prognostic value of CAF expression in PeCa. There is only one study on PeCa and CAF. The study conducted by Cury et al. on 63 PeCa samples elucidates the critical roles of immune cells and cancer-associated fibroblasts (CAFs) in creating a TIME through the expression and potential secretion of inflammatory factors and ECM remodeling proteinases. The observed negative correlation between immune cell proportions and CAFs in PeCa samples suggests a complex interplay between these cellular components within the TIME. Furthermore, the analysis demonstrated that patients with elevated CAF scores exhibited reduced survival rates, indicating the prognostic significance of CAFs in PeCa. This association was accompanied by an increased expression of matrix metalloproteinases and collagens in high-CAF samples. These findings indicate the potential role of CAFs in promoting tumor progression and metastasis through ECM remodeling and the creation of a more permissive environment for cancer cell growth and invasion [66]. Given the current lack of comprehensive data regarding the role of CAFs in PeCa, future research should prioritize an in-depth characterization of CAFs within this context. This approach would provide a background for understanding the specific biological roles of CAFs in PeCa, thereby enabling more focused research and the development of potential therapeutic interventions. Furthermore, investigating the signaling pathways influenced by CAFs will clarify the mechanisms by which CAFs contribute to the progression and metastasis of PeCa. Investigating these research domains enables researchers to gain a deeper understanding of CAFs in PeCa, which may facilitate the development of prognostic tools and therapeutic strategies.

## 3. Molecular Basis of Penile Cancer and the Inflammatory Process

The molecular basis for the development of PeCa includes a distinction between cancers that develop as a result of HPV infection, known as HPV-dependent (or positive) cancers, and non-HPV-related cancers, known as HPV-independent (or negative) cancers [67]. HPV 16 is by far the most common genotype in penile cancer [68]. A meta-analysis by Vaanthoor et al. showed that nearly 51% of PeCa cases were associated with HPV infection (n = 4199), which develops inside epithelial cells and, through the integration of viral DNA into the host cell genome, leads to the production of viral proteins that interact with the cell cycle of human cells as oncoproteins [69].

The viral proteins E7 and E6 interfere with cell-cycle regulation; E7 inactivates the pRB protein, leading to the release of the transcription factor E2F, which drives further steps in the cell cycle. In contrast, E6 blocks the suppressor protein TP53, leading to the inhibition of cell-cycle blockades and apoptosis. The over-expression of the cycle inhibitor p16INK4A in mucosal cells, detected by immunohistochemistry, indicates active HPV development and is a diagnostic parameter in PeCa [70].

The molecular basis of HPV-negative penile cancer is more complex, as several cellular pathways may be involved simultaneously or sequentially. The uncoupling of elements of pathways related to the cell cycle and/or apoptosis, i.e., cyclin D/RB protein or MDM/p53, has been observed in this type of cancer [69]. Reduced expression of the p16INK4A gene due to DNA lesions (LOH or hypermethylation of genomic DNA), resulting in the absence of the p16INK4A protein, has been reported [70]. Mutation of the TP53 gene may result in the absence or increased levels of mutant TP53 protein [71]. Other authors have observed deregulation of the mTOR pathway in PeCa via the increased expression of mTOR kinase and/or its activators (EGFR, HER3, and HER4) [72,73]. In an induced mouse model, it was shown that when the Apc and Smad4 genes were inactivated, the development of SCC HPV-negative PeCa was observed in 100% of cases. From the point of view of the inflammatory process, it is noteworthy that HPV-negative PeCa are characterized by increased levels of cyclooxygenase-2 in both early and advanced PeCa cells [74]. This leads to the overexpression of prostaglandins and thromboxanes, which activate a number of cellular pathways related to cell survival and proliferation, as well as angiogenesis processes (via VEGFC secretion) [74,75].

The impact of the molecular basis of PeCa has important implications for patient survival. In contrast to cervical cancer, where the presence of HPV is a negative prognostic factor, it has a positive influence on patient survival in external genital cancers. In vulvar cancer, improved overall survival (OS) and disease-free survival (DFS) have been observed for HPV-positive cancers. Similarly, in penile cancer (PeCa), multiple large-scale studies have confirmed the favorable impact of HPV on survival and hazard ratios [76,77,78,79]. Also, in other cancers where the presence of HPV is one of the key molecular underpinnings, such as HNSCC, a better OS and HR of 0.42 has been reported for HPV-positive cases [80]. A meta-analysis of 50 patients from Norway (n = 277) by Moen et al. showed a better CSS in advanced PeCa (pTN1-2M) with HPV-positive status (HR = 0.54) compared to HPV-negative PeCa tumors [81]. The association between the presence of HPV and better patient prognosis in SCC may be related to the activation of TIME cells, leading to increased activation of the immune response against HPV-infected cells compared to HPV-negative tumor cells. Of fundamental importance is the expression of viral proteins, which are antigenically foreign to the organism and lead to increased activation of the immune system, mainly TILs and macrophages [82]. Another important indicator is the tumor mutational burden (TMB), which measures the frequency of DNA mutations in tumor cells. TMB is important in determining the likely efficacy of immunotherapy in a tumor, but is not clinically relevant in the case of PeCa [69]. In this cancer, low TMB levels were found, indicating a low frequency of protein gene mutations and thus a reduced activation of immune cells in terms of inflammatory activation [83]. On the other hand, a study by Necchi et al. on a group of 397 PeCas showed that advanced HPV-positive PeCas were characterized by high TMB levels (c.a. TMB ≥ 10 mut/Mb), and the proportion of HPV-negative cancers was low. The main genes mutated in these cases were PIK3CA, KMT2D, and CDKN2A, which may indicate the novel pathways activated by HPV in PeCa [83].

Due to the rarity of PeCa, studies of the TIME in relation to HPV status have been performed in another SCC, HNSCC [82]. Increased numbers and activity of TILs in HPV-positive HNSCC tumors, mainly CD8+ T lymphocytes, are associated with better patient responses to treatment [84,85]. The increased infiltration of CD8^+^ lymphocytes, both within the tumor and in the stroma, is positively associated with OS in HPV-positive patients. A subpopulation of CD4^+^ cells with a CD161^+^ phenotype specific for HPV-infected cells has been reported to lead to improved OS in HNSCC [84]. HPV-positive SCC cells in both HNSCC and PeCa are characterized by increased levels of PD-L1 and MHC class I and II molecules compared to HPV-negative tumors, which when bound to the PD-1 receptor on the surface of CD8^+^ T cells cause lymphocyte exhaustion [82]. Next, HPV-specific CD8^+^ and CD4^+^ T cells participate in dendritic cell recruitment and tumor inflammation through the production of pro-inflammatory cytokines, such as TNF-α, IFN-γ, and IL-17A. However, CD4^+^ cells and Tregs suppress inflammation through the production of IL-4, IL-5, and TGF-β, respectively [86]. Interestingly, CD8^+^ and FOXP3^+^ cells are more abundant in the stroma than in the tumor tissue, regardless of human papillomavirus (HPV) status. Nevertheless, in HPV-positive TIMEs, more CD8^+^ cells are observed than FOXP3^+^ cells, suggesting a more robust immune response. This phenomenon could potentially explain the better prognosis observed in individuals diagnosed with HPV-positive PeCa [11,87]. However, Lohse et al. suggested that categorizing PeCas as HPV-positive or HPV-negative is insufficient for predicting prognosis. A study of PeCa cell lines and tissue microarrays revealed that PeCa specimens characterized as HPV^+^, p63^+^, CD15^+^, DKK1, and CD147^+^ were associated with more aggressive and metastatic tumors. This mechanism is likely related to the elevated membranous expression of CD147, which suppresses the neutrophil-mediated killing of tumor cells in response to high viral oncoprotein expression [88]. Furthermore, Guimarães et al. reported a high prevalence of HPV infections (66.6%) in a cohort of 30 PeCa cases. Significantly, co-culturing lymphocytes with tumor supernatants from these patients resulted in a reduction in apoptosis, suggesting that HPV plays a potential role in modulating the TIME [89].

Other researchers have found an increased accumulation of B lymphocytes in HPV-positive PeCa, even to the point of forming lymph nodes (tertiary lymphoid structures). This is linked to the presence of viral proteins on the surface/inside of tumor cells and the production of antibodies by B cells. These antibodies bind to membrane proteins of infected SCC cells and activate other cells of the immune system (Figure 1). Therefore, a positive effect of increased B-cell accumulation on OS was observed [90,91]. On the other hand, there were no differences in the number of CD163^+^ M2 macrophages depending on HPV status in PeCa, while the effect of TAMs on tumor progression was described above [11].

## 4. The Role of Pro-Inflammatory Cytokines and Chemokines in Penile Cancer Progression and Prognosis

Cytokines are small proteins that are mainly secreted by macrophages, lymphocytes, NK cells, fibroblasts, keratinocytes, and endothelial cells. The cells primarily responsible for cytokine production in the TIME are shown in Figure 1. They play a role in autocrine, paracrine, and endocrine signaling between cells and tissues. These proteins primarily regulate the proliferation of immunoregulatory cells either by promoting a pro-inflammatory response or by suppressing the anti-inflammatory response. Additionally, they can affect targeted cell migration through chemotaxis—the movement of cells, including leukocytes—in response to chemical signals. Cytokines can exhibit pleiotropic, antagonistic, or synergistic effects, and an equilibrium between the opposing actions of pro-inflammatory and anti-inflammatory cytokines is crucial for sustaining tissue wellbeing [92,93,94,95]. The microenvironment of tumors can be classified into two categories: inflamed and non-inflamed. The differences between these two categories are substantial and are primarily influenced by the mixture of immunological elements that they contain. Pro-inflammatory cytokines, such as interleukin-1 alpha (IL-1 α), interleukin-1 beta (IL-1β), interleukin-2 (IL-2), interleukin-12 (IL-12), interleukin-23 (IL-23), tumor necrosis factor alpha (TNF-α), and interferon (INF) types I and II, are the primary constituents of the inflammatory microenvironment. This type of immune microenvironment is conducive to T-cell activation and expansion. In contrast, the non-inflamed microenvironment is characterized by cytokines that promote immune suppression or tolerance, such as interleukin-4 (IL-4), interleukin-10 (IL-10), interleukin-13 (IL-13), interleukin-17 (IL-17), programmed death ligand 1 (PD-L1), and transforming growth factor beta (TGF-β) [6,11,96] (Figure 2). Current research on pro-inflammatory cytokines explains that IL-1A, IL-1B, IL-6, TGF-β1, and IFN-γ might be key components of the PeCa TIME and are believed to play a significant role in cancer formation [43]. Consequently, these cytokines may contribute to an inflamed PeCa TIME. Conversely, several studies confirm the higher expression of PD-L1 in PeCa, which contributes to immune suppression and tolerance, thereby potentially creating a non-inflamed TIME [97] (Figure 2).

### 4.1. IL-1 Family

IL-1A, IL-1B, and interleukin-1 receptor antagonist (IL-1Ra) are cytokines secreted by macrophages, keratinocytes, and endothelial cells, and they play a role in inflammatory and fibrotic processes. Additionally, IL-1 plays a significant role in tumor growth and metastasis through its diverse biological activities [98]. The IL-1 family comprises three members: IL-1A and IL-1B, which are active, and IL-1Ra, which functions as an antagonist. Although encoded by separate genes, IL-1A and IL-1B exhibit identical biological functions, promoting inflammation and tumor progression. In contrast, IL-1Ra binds to the IL-1 receptor without inducing signal transduction, effectively blocking the biological response typically associated with IL-1. The IL-1 signaling pathway involves two receptors from the immunoglobulin superfamily: interleukin 1 receptor type 1 (IL-1R1) and interleukin 1 receptor type 2 (IL-1R2). IL-1R1 is ubiquitously expressed, biologically active, and responsible for transmitting the IL-1 signal. Conversely, IL-1R2 functions as an IL-1 scavenger, regulating the availability of the cytokine. When IL-1 binds to IL-1R1, it induces the recruitment of the IL-1 receptor accessory protein (IL-1RaP), forming a heterodimeric complex. The formation of this complex initiates the IL-1 signaling cascade, resulting in the activation of various transcription factors. These transcription factors subsequently regulate the expression of genes involved in inflammation, cell proliferation, and survival, thereby contributing to the pro-tumorigenic effects of IL-1 [99,100]. Interleukin-1 plays a significant role in tumor growth and metastasis by promoting metastasis and angiogenesis through the induction of various genes and factors. Specifically, IL-1 upregulates matrix metalloproteinases (MMPs), which facilitate the degradation of extracellular matrix components, allowing cancer cells to invade surrounding tissues and enter the bloodstream. Concurrently, IL-1 enhances the expression of endothelial adhesion molecules, which aid in the attachment of circulating tumor cells to blood vessel walls, a critical step in the metastatic cascade. These combined effects make a significant contribution to the dissemination of cancer cells to distant sites within the body. In addition to its pro-metastatic effects, IL-1 is a potent inducer of angiogenesis, the formation of new blood vessels. This process is essential for tumor growth and metastasis, as it provides the necessary oxygen and nutrients to sustain rapidly dividing cancer cells. IL-1 achieves this by stimulating the production of various pro-angiogenic factors, including VEGF, chemokines, and prostaglandin E2 (PGE2). Furthermore, IL-1 promotes the release of growth factors and transforming growth factor-beta (TGF-β), which not only support angiogenesis but also contribute to tumor progression by modulating the tumor microenvironment and suppressing antitumor immune responses [100,101,102,103]. The combined action of these IL-1-induced factors creates a favorable environment for tumor growth, invasion, and metastasis. According to León et al., in a study that enrolled 154 patients with head and neck cancer, IL-1A overexpression was associated with a higher risk of distant metastasis [104]. The study conducted by Chen et al. found that the expression of IL-1B was elevated in esophageal squamous cell carcinoma (SCC) compared to nonmalignant tissues. To elucidate the significance of IL-1B in esophageal squamous cell carcinoma, investigators examined its effects on tumor cell growth, invasion, and treatment responses through the regulation of IL-1B signaling. In vitro experiments demonstrated that inhibiting IL-1B signaling with an IL-1B antibody significantly reduced tumor cell growth and invasion abilities. Furthermore, the inhibition of IL-1B signaling enhanced cell death induced by cisplatin and irradiation, suggesting that IL-1B may play a crucial role in tumor behavior and treatment resistance. To assess the clinical relevance of these findings, the predictive power of IL-1β on clinical outcomes and treatment response in patients with esophageal SCC was evaluated using immunohistochemistry (IHC) analysis. Moreover, to evaluate the clinical significance of these findings, the predictive capacity of IL-1B regarding clinical outcomes and treatment responses in patients with esophageal SCC was assessed utilizing immunohistochemistry (IHC) analysis. Additionally, positive immunohistochemistry (IHC) staining for IL-1B exhibited a significant association with a reduced response to neoadjuvant chemoradiotherapy and inferior clinical outcomes [105]. Similarly, in a study conducted by Elaraj et al., the IL-1 mRNA exhibited elevated expression in metastatic samples and cancer cell lines derived from patients with non-small-cell lung carcinoma, colorectal adenocarcinoma, and melanoma [106].

### 4.2. IL-6

IL-6 is a multifunctional cytokine that plays a crucial role in the immune system and various physiological processes. Initially identified as a B-cell stimulatory factor, IL-6 has since been recognized for its diverse range of biological activities. This 26-kD secreted protein not only promotes antibody production in B cells, but also influences T-cell differentiation, hematopoiesis, and acute-phase protein synthesis in the liver. The pleiotropic nature of IL-6 is further exemplified by its involvement in inflammation, metabolism, carcinogenesis, and neural processes [95]. The IL-6 signaling pathway is complex and involves multiple components, including the IL-6 receptor (IL-6R) and the signal transducer glycoprotein 130 (gp130). IL-6 can activate cells through two distinct mechanisms: classical signaling and trans-signaling. In classical signaling, IL-6 binds to membrane-bound IL-6R, which then associates with gp130 to initiate intracellular signaling cascades. Trans-signaling, conversely, involves a soluble form of IL-6R that can bind IL-6 and activate cells that do not express membrane-bound IL-6R. This dual signaling capability enables IL-6 to exert its effects on a wide range of cell types and contributes to its diverse biological functions. The overlapping activities of IL-6 with other members of the IL-6 family of cytokines, which also utilize gp130 for signal transduction, further underscore the complexity and significance of this signaling network in maintaining homeostasis and mediating various physiological responses [95,107]. Gilabert et al. revealed a significant correlation between elevated levels of IL-6 expression and poor prognosis in pancreatic cancer patients. Their worsened prognosis was attributed to the promotion of cancer development accompanied by cachexia [108]. The elevated levels of IL-6 in serum have been identified as a significant factor in the progression and prognosis of colorectal cancer. This pro-inflammatory cytokine plays a crucial role in promoting the epithelial-to-mesenchymal transition (EMT), which contributes to the loss of cell–cell adhesion and increased motility, allowing cancer cells to detach from the primary tumor site and invade the surrounding tissues. As a result, elevated IL-6 levels contribute to enhanced tumor invasiveness and metastatic potential in colorectal cancer patients. Furthermore, increased serum IL-6 levels have been correlated with larger tumor sizes and a higher likelihood of metastasis in colorectal cancer. Consequently, these factors contribute to decreased survival rates among colorectal cancer patients [109]. Similarly, a study of 85 patients with muscle-invasive bladder cancer revealed that IL-6 was expressed at higher levels in bladder cancer compared to non-malignant tissues. Furthermore, positive staining for IL-6 was primarily associated with muscle-invasive bladder cancer relative to lower-stage Ta–T1 disease. Additionally, urinary levels of IL-6 were significantly elevated in patients with locally advanced bladder TCC compared to patients with NMIBC. Consequently, IL-6 expression may be associated with a more malignant phenotype [110].

### 4.3. TGF-β

The TGF-β family plays a crucial role in regulating diverse cellular processes and has significant implications for organism homeostasis and disease. This family of proteins, which includes TGF-β, activins/inhibins, and bone morphogenic proteins (BMPs), exerts its effects through a complex signaling cascade. When TGF-β ligands bind to their receptors, they initiate a series of phosphorylation events, ultimately leading to the activation of SMAD proteins. These activated SMADs translocate to the nucleus, where they regulate the transcription of target genes, influencing various cellular functions, such as proliferation, apoptosis, differentiation, EMT, and migration [111,112,113]. The role of TGF-β in cancer development is particularly noteworthy due to its context-dependent effects. In the early stages of carcinogenesis, TGF-β functions as a tumor suppressor by inhibiting cell-cycle progression and promoting apoptosis. However, as cancer progresses, TGF-β undergoes a functional transition, becoming a promoter of tumor growth, invasiveness, and metastasis. This dual nature of TGF-β signaling underscores the complexity of its role in cancer biology. Furthermore, the TGF-β pathway interacts with other signaling pathways in both synergistic and antagonistic ways, contributing an additional layer of complexity to its regulatory functions. The clinical significance of TGF-β activity is emphasized by the association between elevated levels of TGF-β and poor prognosis in various cancers, rendering it an important target for therapeutic interventions and biomarker development in oncology [114,115]. A study conducted by Hegele et al. compared plasma levels of latent and active TGF-β1 in patients with localized RCC (n = 39) and metastatic RCC (n = 17), with benign diseases as a control group (n = 93). No significant differences in TGF-β1 levels were observed between localized RCC patients and the control group. Similarly, no significant variations in TGF-β1 levels among localized RCC subgroups based on the cancer stage and grade were identified. However, significantly higher levels of both latent and active TGF-β1 were found in metastatic RCC patients compared to those with localized RCC. These findings suggest a potential association between elevated TGF-β1 levels and advanced stages of RCC [116]. The research conducted by Chen et al. revealed a notable link between the Int7G24A (rs334354) intronic variation in the TGFBR1 gene and an elevated susceptibility to both bladder transitional cell carcinoma (TCC) and RCC. Individuals with heterozygous or homozygous single-nucleotide polymorphisms (SNPs) at this locus showed a higher susceptibility to these urological cancers. Additionally, a somatic mutation causing a serine-to-phenylalanine substitution at codon 57 of TGFBR1 was discovered. These findings suggest a common genetic factor in bladder and kidney cancer predisposition and highlight the potential role of TGF-β signaling pathway alterations in cancer pathogenesis [117]. Fang et al. revealed miR-34b to be a crucial regulator of the TGF-β pathway in prostate cancer cells. The miR-34b modulates the expression of TGF-β, TGFBR1, pSMAD4, and p53. The upregulation of miR-34b in PC3 prostate cancer cells inhibits cell growth, migration, and invasion. These findings suggest that targeting miR-34b could be a promising therapeutic approach for prostate cancer treatment due to its ability to affect multiple aspects of cancer progression through TGF-β signaling modulation [118].

### 4.4. IFN-γ

Interferon-γ (IFN-γ) is a cytokine with diverse biological functions, particularly in immune regulation and host defense against pathogens. As a member of the interferon family, it was initially identified for its capacity to interfere with viral replication. IFN-γ is unique among interferons, being the sole representative of type II interferons. It is produced by various immune cells, including both innate and adaptive immune components, in response to potentially harmful stimuli. This cytokine plays a vital role in numerous physiological processes, extending beyond its primary functions in immune modulation and antimicrobial defense. The impact of IFN-γ extends to various aspects of human health and disease. It is involved in pregnancy, obesity, allergic reactions, and autoimmune disorders [119,120,121]. IFN-γ has been the subject of extensive research in cancer biology, revealing its complex and occasionally contradictory roles. Initially, it was recognized as a potent antitumor agent. However, subsequent studies have uncovered a more nuanced perspective on the influence of IFN-γ on cancer development and progression. While IFN-γ can inhibit tumor initiation and growth, it also plays a role in shaping tumor immunogenicity. Paradoxically, it can promote the emergence of tumor cells with enhanced capabilities to evade immune detection and elimination [121,122]. A comprehensive bioinformatics study utilizing the Cancer Genome Atlas Program (TCGA) database identified CDKN3 as a potential prognostic indicator in ccRCC. Elevated CDKN3 levels correlate with reduced overall survival and unfavorable outcomes in clear cell RCC patients. This effect is partly mediated through the activation of inflammatory pathways, including IL-6/JAK/STAT3, TNF-α/NF-kB, and IFN-γ signaling [123]. In contrast, Otessen et al. investigated the effects of recombinant human interferon γ (rHu-INFγ) on malignant and pre-malignant urothelial cell lines. Their findings revealed that malignant cells experienced significant growth inhibition (>50%) when exposed to rHu-INFγ, while pre-malignant cells showed less sensitivity. The study suggests that rHu-INFγ has potential as a treatment for human bladder cancer, particularly for malignant urothelial cells [124].

Research on the expression of pro-inflammatory cytokines in PeCas is limited. However, a preliminary study found elevated expression levels of genes encoding IL-1A, IL-1B, IL-6, TGF-β1, and IFN-γ in PeCa tissues. Additionally, a positive correlation was observed between TNM stage and IFN-γ levels, suggesting a dual role of IFN-γ. Although a higher expression level of IFN-γ may indicate an antitumor response, it could also enhance tumor immunogenicity and promote the growth of tumor cells with immunoreactive properties. However, no significant relationship was observed between the expression of pro-inflammatory cytokine-encoding genes and prognosis in PeCa. Nevertheless, it should be noted that the cited study was preliminary in nature, involving only 6 patients with PeCa and 13 controls [121,125]. Moreover, Zhou et al. investigated PeCa cell lines and found genomic changes in the TGF-β pathway, suggesting a potential role of this pathway in cancer development [126]. The primary clinical implication derived from the cited research is that pro-inflammatory cytokines may serve as potential targets for immunotherapy or anti-inflammatory treatments. Furthermore, the observed positive correlation between TNM stage and IFN-γ levels suggests that IFN-γ could function as a biomarker for disease progression. However, due to the current insufficiency of data, further studies are warranted. Future research should focus on larger-scale studies involving more PeCa patients to validate the preliminary findings regarding cytokine expression levels and their correlation with PeCa progression. Additionally, it is essential to investigate the specific mechanisms by which pro-inflammatory cytokines contribute to PeCa development and progression. Exploring these research areas could establish pro-inflammatory cytokines as prognostic biomarkers and targets for therapeutic agents.

### 4.5. Inflammasomes

Inflammasomes are complex protein structures that serve as critical components of the innate immune system, acting as intracellular sensors and initiators of inflammatory responses. They are mediated by proinflammatory cytokines, including IL-1β and IL-18, among others [127]. These structures are composed of multiple proteins, including a sensor protein (such as NLR family pyrin domain containing 3 (NLRP3), absent in melanoma 2 (AIM2), or NLR family CARD domain-containing protein 4 (NLRC4)), an adaptor protein (apoptosis-associated spect-like protein containing a CARD (ASC)), and an effector protein (pro-caspase-1). When activated by various stimuli, including pathogen-associated molecular patterns (PAMPs), damage-associated molecular patterns (DAMPs), or cellular stress signals, these components assemble to form the active inflammasome complex [128]. This process initiates a series of reactions that ultimately result in caspase-1 activation, a crucial enzyme in inflammation. The inflammasome-mediated activation of caspase-1 has widespread implications for immune responses and cellular functions. Caspase-1 cleaves pro-inflammatory cytokines, particularly IL-1β and interleukin-18 (IL-18), into their active forms, which are then released from the cell to propagate inflammation and recruit additional immune cells to the site of infection or injury. Furthermore, caspase-1 catalyzes the cleavage of gasdermin D (GSDMD), a protein that, upon activation, forms pores in the cell membrane. These pores not only facilitate the release of mature cytokines, but also result in a rapid form of cell death known as pyroptosis [127,129]. Pyroptosis is characterized by cellular swelling, membrane rupture, and the release of cellular contents, including additional DAMPs, which further amplify the inflammatory response [130]. The ability of inflammasomes to initiate both cytokine release and pyroptosis highlights their central role in regulating innate immune responses, demonstrating their importance in various physiological and pathological processes, including infection control, autoimmune diseases, and inflammatory disorders [127,130].

In the literature, only two studies have investigated the overexpression of inflammasome components in PeCa, including NLRP3 [131] and AIM2 [132].

The NLRP3 inflammasome exhibits a complex role in cancer, with conflicting evidence presented in the literature. Generally, NLRP3 overexpression is associated with poorer outcomes [133]. Research has demonstrated that the NLRP3 inflammasome promotes breast cancer progression and metastasis by triggering the release of IL-1β [134]. Furthermore, the NLRP3 inflammasome plays a crucial role in regulating inflammation in prostate cancer. It contributes to the growth, survival, migration, and invasion of tumor cells by influencing autophagy, mitochondrial metabolism, and EMT [135]. However, some studies have emphasized the protective role of NLRP3 in tumorigenesis. According to Wei et al., NLRP3 expression is upregulated in inflammatory liver tissue but downregulated in cancerous tissue, suggesting that NLRP3 deficiency may contribute to hepatocellular carcinoma progression [136]. A study conducted by Casanova-Martín et al. provided significant insights into the expression of NLRP3 in PeCa specimens. Their findings demonstrated a correlation between NLRP3 expression levels and tumor differentiation. Specifically, poorly differentiated PeCa tumors exhibited elevated NLRP3 expression compared to moderately and well-differentiated tumors or verrucous carcinoma. This observation suggests that NLRP3 may play a crucial role in the progression of PeCa, potentially serving as a marker for more aggressive forms of the disease. This association may also indicate that NLRP3 contributes to the development of more aggressive phenotypes in PeCa [131].

The AIM2 inflammasome, similarly to NLRP3, demonstrates an influence on cancerogenesis, although conflicting data exist. Some studies provide information about AIM2’s involvement in cancerogenesis. Farschian et al. reported that AIM2 expression is notably elevated in both primary and metastatic cutaneous SCC cell lines when compared to normal keratinocytes [137]. In contrast, a study by Dihlmann et al. involving 476 colon cancer specimens demonstrated that the absence of AIM2 expression was associated with decreased survival rates, quicker disease recurrence, and metastatic progression. Furthermore, patients whose tumor cells exhibited a complete absence of AIM2 expression faced a mortality risk from disease progression that was more than threefold higher than those whose tumor cells expressed AIM2 [138]. The study conducted by Tan et al. identified 22 upregulated genes in PeCa tissues, with B-cell lymphoma 2-related protein A1 (BCL2A1) and AIM2 primarily associated with cellular proliferation. These findings were corroborated by elevated mRNA levels of BCL2A1 and AIM2 in tumor samples compared to normal tissue. The investigators determined that BCL2A1 and AIM2 are reliable oncogenes in PeCa, with their overexpression correlated with cancer-specific survival (CSS). The combined expression levels of both genes had a significant influence on CSS, with the double-negative group exhibiting the highest five-year CSS rate (80.5%), followed by the single-positive group (68.6%), and the double-positive group (38.9%). Notably, the presence or absence of AIM2 did not affect the secretion of cleaved IL-1β and IL-18 in PeCa cells, suggesting that AIM2 may not depend on antitumor inflammatory cytokines to influence immune responses in PeCa cells [132].

### 4.6. Chemokines

Chemokines are a specific type of cytokine that are responsible for attracting leukocytes in response to pro-inflammatory cytokines and growth factors. Altered chemokine expression has been observed in various types of cancers, including breast, prostate, pancreatic, and PeCa. Some studies have linked the increased expression of chemokines such as C-X-C motif ligand 5 (CXCL5), C-X-C motif ligand 13 (CXCL13), and C-C motif ligand 20 (CCL20) to PeCa. Additionally, researchers are investigating the potential association between higher chemokine expression and patient outcomes in PeCa [139,140,141,142].

CXCL5 is a chemokine secreted by immune cells such as monocytes and T lymphocytes. Studies have shown that CXCL5 is overexpressed in over 14 types of cancers, including hepatocellular carcinoma, prostate cancer, pancreatic cancer, and stomach cancer. A significant upregulation of the CXCL5 gene has been observed in PeCa tissue compared to healthy tissue, leading to further investigation of the potential relationship between serum levels of CXCL5 in patients diagnosed with PeCa. Miao et al. reported that patients with PeCa have significantly elevated CXCL5 serum levels. Furthermore, it has been demonstrated that, after PeCa resection, the levels of this chemokine significantly decrease. Patients with higher preoperative CXCL5 serum levels were found to have more advanced disease stages, more lymph node metastases, and shorter survival times [142].

CXCL13 is a chemokine that directs B lymphocytes to the spleen and lymph nodes. Research has shown that CXCL13 is expressed in various types of cancers, including gastric, colorectal, breast, prostate, oral, lung, and hepatocellular carcinomas. In a study conducted by Miao et al., higher expression levels of CXCL13 mRNA were observed in PeCa tissues than in healthy tissues. Additionally, the study found that serum CXCL13 levels were elevated in patients with PeCa and decreased following surgical resection. Other studies have shown a significant relationship between CXCL13 levels and tumor stage, as well as the presence of pelvic lymph node metastasis. However, no correlation was observed between CXCL13 levels and HPV infection status. Collectively, these findings suggest that CXCL13 can be considered a diagnostic and prognostic biomarker for HPV-negative PeCa [143].

CCL20 is involved in the recruitment of monocytes, CD4^+^, and dendritic cells (DCs) to damaged tissues or sites of infection. Research has shown that PeCa tissues express higher levels of CCL20 mRNA than healthy controls. Additionally, serum CCL20 levels were higher in patients with PeCa than in the controls. Notably, a reduction in CCL20 serum levels was observed in patients following the surgical removal of cancer. Moreover, elevated CCL20 levels are correlated with an advanced tumor stage and lymph node metastasis. Nevertheless, no statistically significant association between CCL20 expression and HPV infection has been identified [144].

## 5. Exploring NF-κB Pathway Activation and Its Implications

Nuclear factor-kappa B (NF-κB) and inhibitors of NF-κB kinase (IKK) are fundamental components of the NF-κB pathway and play crucial roles in various physiological processes, including inflammatory responses, cellular differentiation, proliferation, programmed cell death (apoptosis), and survival. The NF-κB gene family comprises five nuclear factors: RelA (p65), c-RelB, c-Rel, NF-κB1 (p50/p105), and NF-κB 2 (p52/p100) [145,146,147]. The formation of various homo- and heterodimers leads to the formation of NF-κB subunit complexes, which in turn regulate the activity of several processes via the NF-κB signaling pathway. Specifically, the p50:p50 homodimer is believed to play a role in anti-inflammatory processes, whereas the p52:p52 complex is thought to be involved in cell proliferation, migration, and inflammation [148,149]. This pathway can be activated in two ways: canonical (NEMO-dependent) and non-canonical (NEMO-independent). The canonical activation of the NF-κB pathway is primarily triggered by pathogen-associated molecular patterns (PAMPs) or pro-inflammatory cytokines, such as IL-1, IL-6, and IL-12, as well as chemokines, including CXCL1 and CXCL2. In contrast, NEMO-independent activation depends solely on TNF receptors, including BAFFR, TNFR2, LTβR, CD40, and RANK. The activation of the NF-κB pathway results in the expression of pro-inflammatory cytokines, angiogenic factors, and other molecules that can promote the survival and proliferation of cancer cells by regulating the genes involved in cell-cycle control and apoptosis [148,150,151,152]. The investigation conducted by Pennatochiotti et al. examined 79 oral SCC patients, categorizing cases by severity. The findings demonstrated elevated NFKB1 mRNA and protein levels in the more advanced stages of oral SCC, suggesting NFKB1’s potential role in cancer progression [153]. Similarly, in a study of 28 patients, Fonseca et al. identified an association between elevated NFKB1 mRNA levels and the severity of oral SCC [154]. The in vitro experiment conducted by Lehman et al. provided valuable insights into the relationship between the canonical NF-κB pathway and metastatic potential in human esophageal epithelial cells. Upon the activation of this pathway, the researchers observed a significant enhancement in the cells’ metastatic capacity. This finding suggests that the NF-κB signaling cascade plays a role in promoting the aggressive behavior of esophageal cancer cells [155].

Limited information is currently available regarding NF-κB expression in PeCa. Wierzbicki et al. reported that NF-κB 1 and NF-κB 2 genes were highly overexpressed in PeCa tissue, indicating the possible presence of NEMO-dependent and NEMO-independent activation pathways. However, it should be noted that only six patients with PeCa were included in that study, and all PeCa tissue samples were HPV-negative [156]. Senba et al. investigated the association between NF-κB and HPV-positive PeCa cases. They found that the percentage of NF-κB positivity in the nucleus or cytoplasm was higher in cases where HPV DNA was present than that in HPV-negative cases [157]. Furthermore, Yang et al. conducted software (MetaCore™) augmented qPCR/DNA kinase active analyses of 11 PeCa cases, which showed the overexpression and upregulation of kinases related to the NF-κB pathway [158]. A study conducted by Casanova-Martín et al. provides significant insights into the role of allograft inflammatory factor 1 (AIF-1) in PeCa. Their findings reveal that AIF-1 is expressed in PeCa tissue samples, suggesting its potential involvement in the disease process. AIF-1 participates in carcinogenesis through its ability to activate key signaling pathways, specifically the NF-κB pathway and β-catenin. Furthermore, the researchers observed a correlation between elevated levels of AIF-1 and more aggressive PeCa tumors. This association suggests that AIF-1 may serve as a potential biomarker for disease severity and progression [131].

## 6. The Secreted Phosphoprotein 1 (SPP1) Gene: From Bone Mineralization to Penile Cancer Prognosis

The secreted phosphoprotein 1 (SPP1) gene is responsible for encoding osteopontin, which primarily functions in the process of bone matrix mineralization. However, some studies have linked SPP1 to tumorigenesis in various cancers, including pancreatic and colorectal cancers, as well as melanoma. In these cancers, the overexpression of SPP1 is associated with poor prognosis [159,160,161]. Interestingly, in a study involving 183 patients with PeCa, SPP1 was overexpressed in tumor and lymph node metastatic tissues. Nevertheless, high SPP1 expression was associated with better prognosis, suggesting that SPP1 may enhance the antitumor immune response by T cells and regulatory T lymphocytes (TILs). Therefore, it is suggested that SPP1 may serve as a prognostic biomarker in patients with PeCa [162].

## 7. C-Reactive Protein as a Biomarker in Cancer: Implications for Penile Cancer Prognosis and Metastasis

C-reactive protein (CRP) is an acute-phase protein that is synthesized by hepatocytes in response to inflammatory stimuli and can be increased in various conditions, such as infections, trauma, autoimmune diseases, or malignancies. CRP production in hepatocytes is mainly controlled by interleukin-6 (IL-6) and interleukin-1β (IL-1β), which activate C/EBP family members, including C/EBPβ and C/EBPδ [163]. Although CRP is commonly associated with infection, it has been suggested that it is associated with various types of cancer, such as breast, colorectal, and lung cancer [164,165,166]. According to Zhu et al., C-reactive protein (CRP) may serve as a potential biomarker for site-specific cancer risk assessment, including head and neck, esophagus, stomach, colorectal, liver, kidney, breast, lung, and non-Hodgkin lymphomas. It should be noted that CRP is likely not the cause of cancer itself, but rather a response marker of environmental risk factors. Several hypotheses have been proposed regarding the potential mechanisms of the association between higher CRP levels and the risk of carcinogenesis, such as inflammation caused by cancer increasing serum CRP levels, the TIME stimulating CRP production, CRP as part of the host immune response to tumor cells, and CRP as a marker of chronic inflammation that could contribute to carcinogenesis [167]. Furthermore, high levels of CRP in the blood are associated with the progression of cancer to advanced stages and poor prognosis [168,169,170] (Figure 3).

Currently, PeCa lacks widely accepted biomarkers for predicting occult lymph node metastasis and determining patient surveillance. To address this knowledge gap, some research has been conducted. According to Al Ghazal et al., who conducted a retrospective, two-center study involving 51 patients with PeCa, elevated preoperative serum CRP levels (24.7 mg/L vs. 6.4 mg/L) were found to be associated with nodal metastasis, potentially aiding in decisions regarding lymph node dissection for patients [171]. Similarly, Steffens et al. showed a significant association between high serum CRP levels (>15 mg/L vs. ≤15 mg/L) and more advanced stages of the disease, inguinal lymph node metastasis, and unfavorable clinical outcomes in patients with PeCa [172]. Li et al. reported that, in their study of 124 Chinese patients with PeCa, elevated levels of both CRP ≥ 4.5 mg/l and squamous cell carcinoma antigen (SCCAg) ≥ 1.4 ng/mL were associated with the pathologic tumor and nodal stage, extranodal extension, pelvic lymph node metastases, and cancer-specific survival (CSS). Additionally, CRP and SCC-Ag levels were found to be independent prognostic factors that could identify patients who would benefit from early inguinal lymph node dissection [173]. Li et al. examined the relationship between CRP levels and body mass index (BMI) and their impact on the prognosis of 172 patients with PeCa. This study found that elevated CRP levels and a lower BMI were independent risk factors for poor CSS [174]. Furthermore, according to Kawase et al., an elevated CRP level was correlated with a poor prognosis for PeCa in a univariate analysis. However, there were no statistically significant differences between cancer-specific survival (CSS) and CRP levels in the multivariate analysis [175]. In contrast, Ghoshal et al. conducted a study of 50 PeCa patients and found no correlation between elevated serum CRP levels and the development of PeCa [176].

## 8. The Neutrophil-to-Lymphocyte Ratio as a Prognostic Biomarker in Penile Cancer

The pursuit of reliable biomarkers of cancer prognosis has gained substantial momentum in recent years. Investigators are focusing on identifying indicators that not only offer predictive value, but also meet practical criteria for clinical implementation. These criteria encompass ease of measurement, reproducibility, and cost-effectiveness. Among the various biomarkers under investigation, inflammatory blood markers have emerged as promising candidates due to their accessibility and potential to reflect the complex interactions between neoplasms and the host immune system [177]. Among other inflammatory biomarkers, the neutrophil-to-lymphocyte ratio (NLR) has garnered particular attention as a potential prognostic factor across multiple cancer types. This ratio, derived from a complete blood count, provides insight into the balance between neutrophils and lymphocytes, which may offer information regarding the patient’s inflammatory status and immune response to the tumor [178].

Neutrophils constitute the main component of white blood cells and are the most prevalent type of granulocytes. These cells are essential to the innate immune response against pathogens, employing various strategies such as phagocytosis, chemotaxis, and the production of granular proteins, reactive oxygen species (ROS), and cytokines. Additionally, neutrophils contribute to adaptive immunity by facilitating the recruitment and activation of diverse immune cells, including T cells, B cells, NK cells, and mesenchymal stem cells during a systemic inflammatory response (SIRS) [179,180]. However, these cells also play a role in carcinogenesis. Neutrophils facilitate tumor initiation through the production of ROS and proteases, resulting in epithelial damage and inflammation. Neutrophils contribute to tumor growth by activating cancer cells, stimulating proliferation, and suppressing antitumor immune responses. Furthermore, neutrophils participate in extracellular matrix remodeling and promote angiogenesis through the activation of VEGFA by matrix metalloproteinase 9 (MMP9) [181].

Lymphocytes, which include T cells, B cells, and NK cells, represent another class of blood cells. These cells are crucial for adaptive immunity, and their activity responds to various stimuli, including viruses, allergens, and cancerous cells [6,182]. An elevated TIL count is frequently associated with improved clinical outcomes, as it indicates an active and robust immune response against the tumor. Conversely, a diminished TIL count may suggest an attenuated or suppressed immune response, potentially facilitating more rapid cancer progression. The detailed mechanism of action of TILs in the TIME is illustrated in Figure 1 and elucidated in the subsection titled *Tumor-Infiltrating Lymphocytes, Macrophages, and Fibroblasts: Key Players in the Cancer Immune Microenvironment.*

A balance between different populations of white blood cells is crucial for maintaining the proper homeostasis of the immune system. The neutrophil-to-lymphocyte ratio (NLR) may be elevated under various conditions, including trauma [183], bacterial or fungal infection [184], stroke [185], atherosclerosis [186], postoperative complications [187], and SIRS [188]. Moreover, the NLR is elevated in different cancer types and may be associated with CSS, PFS, OS, and DFS [189]. Kao et al. investigated the NLR in 173 patients diagnosed with malignant mesothelioma who were undergoing systemic therapy. An elevated NLR was determined to be an independent predictor of poor survival in malignant mesothelioma patients receiving systemic therapy, irrespective of whether this treatment was administered as an initial or subsequent therapeutic intervention [190]. A retrospective study conducted by An et al. included 95 patients with pancreatic cancer. The study found that an elevated NLR was independently associated with shorter survival times in patients with advanced pancreatic cancer [191]. Cedrés et al. conducted a study involving 171 patients diagnosed with stage IV non-small-cell lung cancer. Among these patients, 60 (35.1%) exhibited an NLR ≥ 5. The elevated NLR was associated with reduced survival durations in patients with advanced non-small-cell lung cancer [192]. In a retrospective analysis conducted by Cordeiro et al., 187 patients who underwent radical nephrectomy for locally advanced nonmetastatic ccRCC were studied over an extended postoperative period (median follow-up period of 48.7 months). The findings revealed that patients with high-histological-grade tumors (Fuhrman grade 3–4) and an elevated NLR (≥4.0) experienced poorer recurrence-free survival (RFS). Conversely, in cases of low Fuhrman grades (1–2), no correlation was observed between NLR and RFS [193].

Consequently, the NLR appears to be a practical and cost-effective prognostic biomarker of inflammation and cancer progression, wherein an increased neutrophil count and decreased lymphocyte count in peripheral blood tests are associated with a reduced antitumor response. However, it should be noted that the optimal cutoff value for NLR remains a subject of some debate, as different studies have employed varying NLR cutoff values [179,189].

The impact of the neutrophil-to-lymphocyte ratio on CSS, PFS, DFS, and OS in patients with PeCa has been investigated in numerous retrospective studies, primarily because of the widespread practice of conducting complete blood counts in patients prior to surgical interventions (Figure 3). Li et al. investigated 228 patients with PeCa after bilateral inguinal lymph node dissection, of whom 52.6% were positive. The preoperative neutrophil-to-lymphocyte ratio was an independent factor affecting DFS and CSS [194]. Similarly, Hu et al. investigated the prognostic value of the NLR in a cohort of 225 patients with PeCa. Patients underwent inguinal lymph node dissection, and 34.2% of them exhibited metastasis. An NLR > 2.94 was associated with decreased OS and PFS. Moreover, an elevated NLR has been correlated with nodal involvement [195]. Azizi et al. enrolled 68 patients with PeCa who had undergone a complete blood count before inguinal lymph node dissection, of whom 72% were positive. NLR ≥ 3 was associated with higher-stage disease, positive lymph nodes, and extranodal extension. Moreover, the OS was significantly worse in a cohort of patients with a higher NLR [196]. A similar cutoff of NLR > 3 was adopted by Jindal et al., who included 69 patients with PeCa, of whom 52.2% had positive dissected lymph nodes, with pN3 (36.2%) being the most prevalent. Patients with an NLR > 3 had inferior CSS, which was significantly associated with lymph node metastasis and a higher T stage [197]. According to Jiao Hu et al., who conducted a study involving 79 patients diagnosed with PeCa who underwent bilateral inguinal lymphadenectomy, an elevated NLR was correlated with advanced tumor grades and an increased incidence of lymph node metastasis [198]. Moreover, Zhou et al. observed a significant correlation between a higher NLR and reduced CSS in a study comprising 114 patients with PeCa who underwent a partial or total penectomy with or without inguinal lymph node dissection [199]. Kasuga et al. investigated the association between the NLR and prognosis in 41 patients with PeCa prior to total penectomy. Patients with NLR ≥ 2.8 exhibited a higher frequency of lymph node metastasis, consequently resulting in poorer CSS and OS [200]. Similarly, Tan et al. enrolled 39 patients with PeCa in their study, wherein the group of patients with NLR ≥ 2.8 exhibited significantly worse CSS than the group with NLR < 2.8. Furthermore, elevated NLR values were associated with a more advanced T stage [201]. Pond et al. conducted a multicenter study exploring the prognostic factors in 140 patients with advanced PeCa prior to first-line chemotherapy. Their results indicated that a higher NLR was significantly associated with poorer OS [202]. In contrast, in a study analyzing 65 patients with PeCa before the second or subsequent systemic treatment, Buonerba et al. found that a higher NLR was not significantly correlated with OS and responses to systemic treatment. However, it should be noted that, in this study, the authors adopted a considerably higher NLR cut-off point of 5, which may have influenced their results [203].

A comprehensive summary of the studies investigating inflammation in PeCa is presented in Table 1.

## 9. Conclusions

In this narrative review, we analyzed the substantial current literature on what is known about the molecular basis of both HPV-positive and HPV-negative PeCa. It is evident that the development, progression, and dissemination of PeCa are all associated with inflammatory processes. The tumor microenvironment, characterized by an abundance of inflammatory cells and pro-inflammatory cytokines, contributes to the accumulation of mutations in the epithelium. Furthermore, it facilitates angiogenesis, cell migration, and metastasis. Consequently, pro-inflammatory immune system components warrant consideration as potential targets for novel anticancer therapeutics.

## Figures and Tables

**Figure 1 ijms-26-02785-f001:**
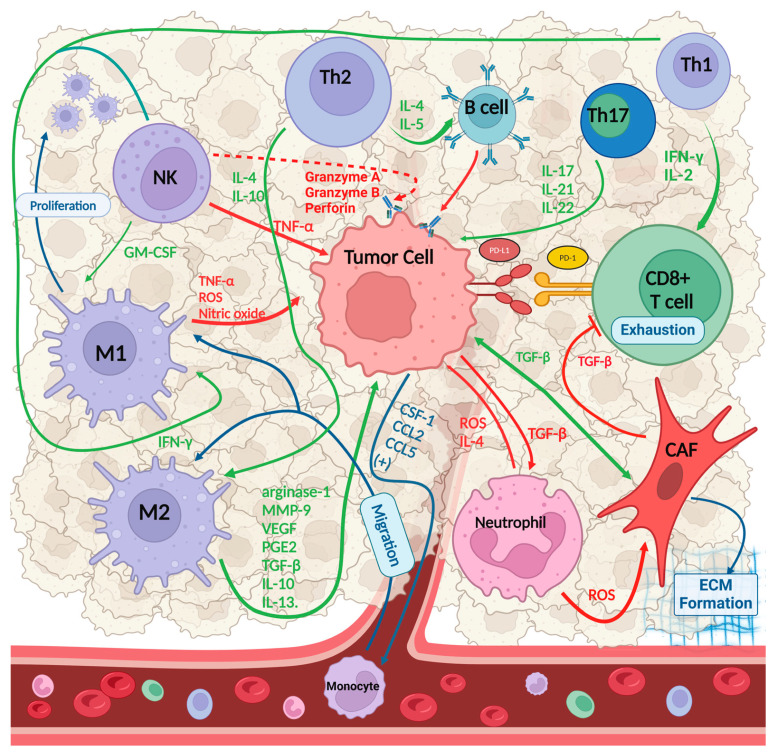
Selected cellular elements of the TIME interacting with PeCa cells. A tumor fragment is shown, along with a blood vessel supplying nutrients and cells from the body. Cellular processes are described in boxes. Abbreviations: M1, M2: TAM populations; NK: natural killer cell; Th1, Th2, Th17: CD8^+^ T-cell populations; CAF: cancer-associated fibroblast. Created using https://BioRender.com (accessed on 23 February 2025).

**Figure 2 ijms-26-02785-f002:**
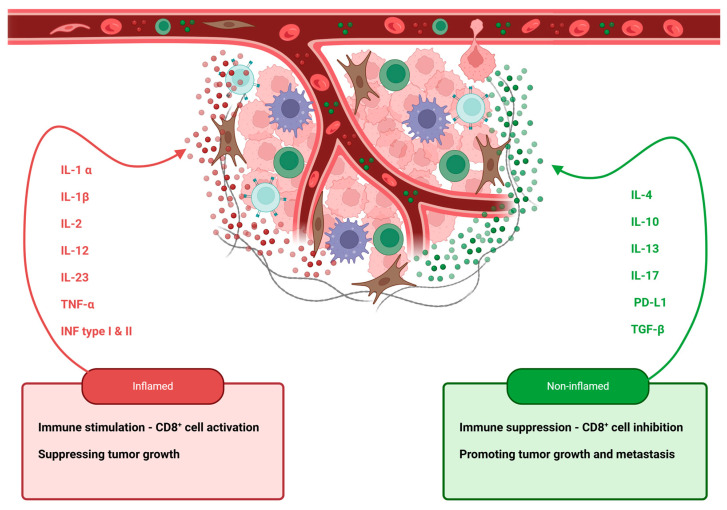
Enhancement or silencing of TIME inflammatory processes in PeCa in cytokine juxtaposition. The presence of pro-inflammatory cytokines (left panel) stimulates CD8^+^ cytotoxic lymphocytes, while the secretion of anti-inflammatory or immunomodulatory cytokines and the over-stimulation of CD8^+^ PDL-1 lymphocytes by cancer cells leads to the inhibition of the activity of these lymphocytes (right panel). Created using https://BioRender.com (accessed on 7 January 2025).

**Figure 3 ijms-26-02785-f003:**
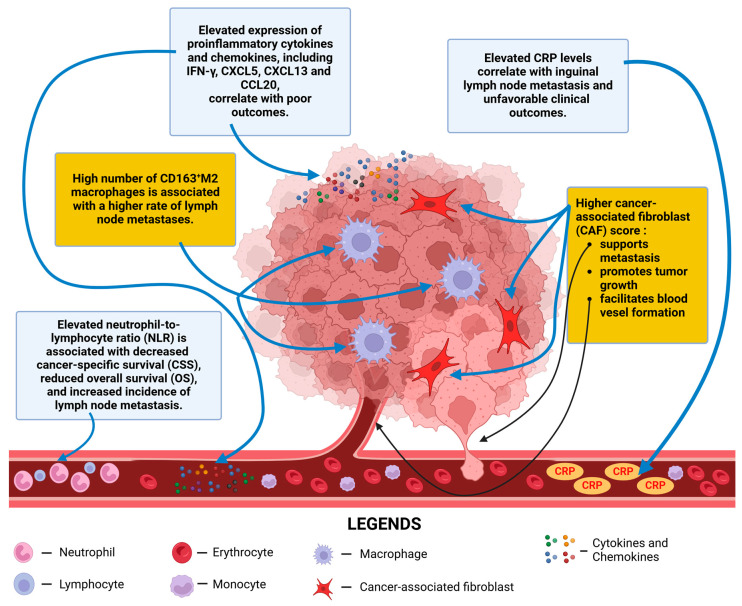
Compilation of inflammation-related elements as prognostic factors in PeCa. The factors shown in blue boxes determine the overall response of the body in the measurement of serum values. Local values analyzed within the tumor are shown in the dark-yellow boxes. Created using https://BioRender.com (accessed on 23 February 2025).

**Table 1 ijms-26-02785-t001:** Inflammatory molecule targets in PeCa.

Author	Particle	Article Type	No. of PeCa	Key Findings	Ref.
**Xu et al.**	TIME	Prospective study	n = 6	Provides insights into the mechanisms driving PeCa progression, premetastatic niche formation, and lymphatic metastasis.	[20]
**Lohse et al.**	TIME	Retrospective study	n = 94	PeCas exhibiting an HPV^+^/p63^+^/CD15^+^/DKK1^+^/CD147^+^ profile are associated with increased aggressiveness and metastasis.	[88]
**Guimarães et al.**	TIME	Prospective study	n = 30	HPV infection may trigger an inadequate immune response, potentially facilitating the development of PeCa.	[89]
**Ottenhof et al.**	TILs	Retrospective study	n = 213	Higher infiltration of CD8^+^ TILs in tumor-associated stroma is associated with lymph node metastasis in PeCa.	[11]
**Vassallo et al.**	TILs	Retrospective study	n = 122	Abundant Fox-P3^+^ cells and pronounced inflammation are significant predictors of poor prognosis in PeCa.	[35]
**Lohneis et al.**	TILs	Retrospective study	n = 28	HPV-associated PeCa exhibits elevated levels of TILs, characterized predominantly by Th1 and cytotoxic profiles. Nevertheless, the increased presence of regulatory T cells (Tregs) in these neoplasms may contribute to immune evasion mechanisms.	[87]
**Hladek et al.**	TILs	Retrospective study	n = 55	PeCa tissues exhibit increased immune cell infiltration, especially CD3^+^, CD8^+^, and CD20^+^.	[36]
**Ottenhof et al.**	TAMs	Retrospective study	n = 213	A high number of intramural CD163^+^ M2 macrophages is significantly associated with a higher incidence of lymph node metastasis. Nevertheless, CD4^+^ T cells could reprogram them into M1.	[11]
**Chu et al.**	TAMs	Retrospective study	n = 178	Elevated levels of CD68^+^ and CD206^+^ TAMs are correlated with a more favorable prognosis.	[6]
**Cury et al.**	CAFs	Retrospective study	n = 63	Patients with elevated CAF scores exhibited reduced survival rates.	[66]
**Czajkowski et al.**	IL-1A, IL-1B, IL-6, INF-γ, and TGF-β	Prospective study	n = 6	Elevated expression of proinflammatory cytokines (IL-1A, IL-1B, IL-6, INF-γ, and TGF-β) in PeCa was observed. A positive correlation was found between higher INF-γ levels and clinical advancement.	[125]
**Zhou et al.**	INF-γ	Retrospective study	n = 114	The IFNγ-mediated induction of IDO1 contributes significantly to the formation of an immunosuppressive tumor microenvironment in PeCa.	[199]
**Casanova-Martín et al.**	NLRP3 inflammasome, AIF-1	Retrospective study	n = 34	Elevated levels of NLRP3 and AIF-1 contribute to the development of more aggressive phenotypes in PeCa.	[131]
**Tan et al.**	AIM2 inflammasome	Retrospective study	n = 220	AIM2 is a reliable oncogene in PeCa, with its overexpression correlated with CSS.	[132]
**Mo et al.**	CXCL5	Retrospective study	n = 81	Elevated preoperative CXCL5 levels predict PeCa progression and may serve as a prognostic biomarker.	[142]
**Mo et al.**	CXCL13	Retrospective study	n = 76	Elevated serum CXCL13 levels correlate with PeCa progression.	[143]
**Mo et al.**	CCL20	Retrospective study	n = 76	Elevated serum CCL20 levels correlate with PeCa progression.	[144]
**Wierzbicki et al.**	NF-κB	Retrospective study	n = 6	Both canonical and non-canonical NF-κB pathways can be activated in PeCa.	[156]
**Senba et al.**	NF-κB	Retrospective study	n = 51	NF-κB was more frequently detected in HPV-positive PeCa.	[157]
**Zou et al.**	SPP1	Retrospective study	n = 183	Elevated SPP1 expression was associated with favorable prognosis in PeCa patients, suggesting that SPP1 may augment antitumor immunity mediated by T cells and regulatory T cells.	[162]
**Al Ghazal et al.**	CRP	Retrospective study	n = 51	CRP could help identify PeCa patients needing inguinal lymph node dissection.	[171]
**Steffens et al.**	CRP	Retrospective study	n = 79	Elevated preoperative CRP levels predicted poor survival in PeCa.	[172]
**Li et al.**	CRP	Retrospective study	n = 124	Combined CRP and SCC-Ag levels predict lymph node metastasis, advanced stage, and survival in PeCa.	[173]
**Li et al.**	CRP	Retrospective study	n = 172	Elevated CRP levels and lower BMI were identified as independent risk factors for poor CSS in PeCa.	[174]
**Kawase et al.**	CRP	Retrospective study	n = 64	High CRP levels were significantly associated with poorer cancer-specific survival (CSS).	[175]
**Ghoshal et al.**	CRP	Retrospective study	n = 50	No association was observed between elevated serum CRP levels and the development of PeCa.	[176]
**Li et al.**	NLR	Retrospective study	n = 228	The preoperative NLR was an independent prognostic factor for both DFS and CSS in PeCa patients.	[194]
**Hu et al.**	NLR	Retrospective study	n = 225	An elevated NLR was associated with decreased OS and PFS. Furthermore, elevated NLR has been correlated with nodal involvement.	[195]
**Azizi et al.**	NLR	Retrospective study	n = 68	An elevated NLR was associated with advanced-stage disease, lymph node involvement, extranodal extension, and significantly reduced OS.	[196]
**Jindal et al.**	NLR	Retrospective study	n = 69	An elevated NLR was correlated with lymph node metastasis, a higher T stage, and inferior CSS.	[197]
**Hu et al.**	NLR	Retrospective study	n = 79	An elevated NLR was associated with advanced tumor grades and increased incidence of lymph node metastasis.	[198]
**Zhou et al.**	NLR	Retrospective study	n = 114	An elevated NLR was correlated with inferior CSS.	[199]
**Kasuga et al.**	NLR	Retrospective study	n = 41	An elevated NLR was associated with a higher incidence of lymph node metastasis and inferior CSS and OS.	[200]
**Tan et al.**	NLR	Prospective study	n = 39	An elevated NLR was associated with higher T stages and significantly worse CSS.	[201]
**Pond et al.**	NLR	Retrospective study	n = 140	An elevated NLR was associated with poorer OS.	[202]
**Buonerba et al.**	NLR	Retrospective study	n = 65	An elevated NLR was not correlated with OS or responses to systemic treatment.	[203]

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
