# Peer review of "Inflammation in Penile Squamous Cell Carcinoma: A Comprehensive Review"

_ijms, 2025, doi:10.3390/ijms26062785_

Round 1
Reviewer 1 Report (Previous Reviewer 1)
Comments and Suggestions for Authors
The revision is appropriate compared to the previous and original version.
No more comments are suggested by the present reviewer.
I recommend the revision for further accept.
Author Response
Thank you for your comment. We are glad that the current version is appropriate.
Reviewer 2 Report (New Reviewer)
Comments and Suggestions for Authors
Dear authors, although I think you did a quality job for the most part, there are some small shortcomings in the manuscript. The text identifies the contradictory roles of specific immune cells, such as B cells and Th17 cells, yet fails to elucidate the underlying mechanisms. The paper addresses prognostic associations of TILs but lacks a thorough examination of their significance for therapeutic interventions. Incorporating clinical perspectives would enhance the overall impact.
The article presumes familiarity with essential concepts including fibroblast activation protein (FAP), epithelial-to-mesenchymal transition (EMT), and paracrine signaling, without providing definitions for these terms.
The text indicates a lack of data concerning the prognostic significance of CAF expression in PeCa, yet it fails to address possible explanations for this deficiency or propose avenues for future research.
The discourse surrounding interleukin families predominantly emphasizes mechanisms, lacking sufficient clinical perspective.
Author Response
We express our gratitude for the comprehensive review of this study. We have endeavoured to address all pertinent questions and revised the manuscript in accordance with the reviewer's recommendations.
Reviewer #2:
- The text identifies the contradictory roles of specific immune cells, such as B cells and Th17 cells, yet fails to elucidate the underlying mechanisms.
Thank you for the comment. The following text has been integrated into the main body of the manuscript within the subsection on tumor-infiltrating lymphocytes.
“…” Th17 cells, through the secretion of distinct cytokines such as IL-17, IL-21, and IL-22, exhibit contradictory effects on carcinogenesis. In colorectal, pancreatic, and non-small cell lung cancers, Th17-related factors are associated with adverse outcomes. Conversely, in breast cancer, IL-22 production is correlated with reduced tumor formation and improved prognosis. Additionally, in ovarian cancer, elevated levels of tumor-associated IL-17 are predictive of improved patient survival. “…”
“…” B cell infiltration is associated with a favorable prognosis in breast and ovarian cancers. Furthermore, research has demonstrated that modulating B cell activity within the tumor immune microenvironment (TIME) can disrupt cancer-induced immunosuppressive events. Conversely, genetic mouse models of skin cancer present contrasting results, indicating a tumor-promoting role for B cells. One potential mechanism may involve the TGF-β-dependent conversion of Forkhead box P3 (FOXP3) cells, which can support and promote metastasis. “…”
- The paper addresses prognostic associations of TILs but lacks a thorough examination of their significance for therapeutic interventions. Incorporating clinical perspectives would enhance the overall impact.
We acknowledge your recommendation. The earlier version of the manuscript included an extensive section on PD-L1 and immune checkpoint inhibitors, supplemented by three additional figures and two tables. However, one of the reviewers did not endorse this content, focusing solely on inflammatory interactions and excluding immunotherapy. Consequently, the manuscript can only briefly acknowledge the existence of PD-L1 and PD-1 without delving deeply into the topic. We have incorporated the information regarding the clinical perspective.
“…” One of the most significant regulatory mechanisms influencing the activity of TILs is the programmed cell death protein 1/programmed cell death ligand 1 (PD-1/PD-L1) axis [37]. PD-1 is a cell surface receptor expressed on various immune cells, including CD8+ T cells, CD4+ T cells, NK cells, monocytes, antigen-presenting cells (APCs), and CD20+ lymphocytes [38]. It plays a crucial role in regulating TILs activity by interacting with its cognate ligand, PD-L1. PD-L1, a membrane-bound protein, forms a complex with PD-1 mainly on CD8+ T cells, leading to the inhibition of immune responses. This mechanism is also employed by cancers as an immune evasion strategy to inhibit the host's antitumor immune response [39]. Consequently, the PD-1/PD-L1 axis is a primary target of immunotherapy in various cancers, including PeCa. The mechanism of immune checkpoint inhibitors (ICIs) generally involves blocking the binding of cancer's PD-L1, thereby facilitating the patient's immune response against cancer [40].
The ongoing clinical trials and studies investigating ICIs for PeCa treatment have shown mixed results, with some promising outcomes but also limitations. The PERICLES Study, a Phase II trial of atezolizumab (anti-PD-L1) with or without radiotherapy, demonstrated a one-year progression-free survival (PFS) rate of 12.5% among 32 participants, with a median overall survival (OS) of 11.3 months and an overall response rate (ORR) of 16.7% [41]. Similarly, the ORPHEUS Study, a Phase 2 trial of retifanlimab (anti-PD-1), reported an ORR of 16.7% in 18 patients, with a median PFS of 2.0 months and median OS of 7.2 months [42]. Other studies have explored combination therapies and different ICIs. A basket trial by Apolo et al. investigated the combination of cabozantinib, nivolumab, and ipilimumab, showing an ORR of 44.4% in nine penile cancer patients, with median PFS and OS of 4.8 and 6.7 months, respectively [43]. A retrospective analysis by Rouvinov et al. reported near-complete response in three patients treated with cemiplimab (anti-PD-1) [44]. Ongoing trials, such as the EPIC trial studying cemiplimab, continue to explore the potential of ICIs in penile cancer treatment [45]. However, some basket trials, including those by McGregor et al. and Naing et al., have shown limited or no response in penile cancer patients [46,47]. These studies highlight the need for further research with larger sample sizes to better understand the efficacy of ICI therapy in penile cancer and identify potential biomarkers for patient selection. Therefore, the primary clinical implication of TILs in PeCa lies in their potential use as a prognostic biomarker. Furthermore, TILs could serve as targets for immunotherapy, either alone or in conjunction with traditional chemotherapy. Moreover, the evaluation of the concentration of specific types of TILs in PeCa may serve as a predictive biomarker for the response to immunotherapy. This assessment facilitates the identification of patients who are more likely to benefit from such treatments. Additionally, understanding the composition and distribution of TILs within the TIME could provide insights into the mechanisms of immune evasion employed by PeCa cells, potentially leading to the development of novel therapeutic approaches. Moreover, longitudinal monitoring of TILs levels during treatment in clinical trials could offer valuable information about the dynamic immune response which may change PeCa patients prognosis. “…”
- The article presumes familiarity with essential concepts including fibroblast activation protein (FAP), epithelial-to-mesenchymal transition (EMT), and paracrine signaling, without providing definitions for these terms.
The authors acknowledge the recommendation and have incorporated the content into the body of the manuscript.
“…” Lopez et al. reported that in clear cell renal cell carcinoma (ccRCC), fibroblast activation protein (FAP), which has collagenolytic activity and is highly expressed on CAFs, was correlated with increased tumor size, higher grade, and more advanced stage. “…”
“…” The epithelial-to-mesenchymal transition (EMT) is a cellular program that transforms epithelial cells into mesenchymal-like cells. Notably, key differences between epithelial and mesenchymal cells include the presence of cell junctions, motility, and invasiveness. Therefore, EMT is associated with increased cell motility and invasiveness, which could contribute to cell separation from the primary tumor and facilitate metastasis. “…”
“…” This paracrine signaling axis which is a form of cellular communication in which a cell emits a signal to induce changes in nearby cells, thereby influencing their behavior, appears to play a significant role in promoting tumor growth and metastasis. “…”
- The text indicates a lack of data concerning the prognostic significance of CAF expression in PeCa, yet it fails to address possible explanations for this deficiency or propose avenues for future research.
The authors have considered the suggestion and incorporated the material into the subsection of CAFs in the manuscript.
“…” Given the current lack of comprehensive data regarding the role of CAFs in PeCa, future research should prioritize an in-depth characterization of CAFs within this context. This approach would provide the background for understanding the specific biological roles of CAFs in PeCa, thereby enabling more focused research and the development of potential therapeutic interventions. Furthermore, investigating the signaling pathways influenced by CAFs will clarify the mechanisms by which CAFs contribute to the progression and metastasis of PeCa. Investigating these research domains enables researchers to gain a deeper understanding of CAFs in PeCa, which may facilitate the development of prognostic tools and therapeutic strategies. “…”
- The discourse surrounding interleukin families predominantly emphasizes mechanisms, lacking sufficient clinical perspective.
The authors acknowledge the recommendation. The subsequent text has been incorporated into the main body of the manuscript within the subsection on cytokines.
“…” The primary clinical implication derived from the cited research is that pro-inflammatory cytokines may serve as potential targets for immunotherapy or anti-inflammatory treatments. Furthermore, the observed positive correlation between TNM stage and IFN-γ levels suggests that IFN-γ could function as a biomarker for disease progression. However, due to the current insufficiency of data, further studies are warranted. Future research should focus on larger-scale studies involving more PeCa patients to validate the preliminary findings regarding cytokine expression levels and their correlation with PeCa progression. Additionally, it is essential to investigate the specific mechanisms by which pro-inflammatory cytokines contribute to PeCa development and progression. Exploring these research areas could establish pro-inflammatory cytokines as prognostic biomarkers and targets for therapeutic agents. “…”
This manuscript is a resubmission of an earlier submission. The following is a list of the peer review reports and author responses from that submission.
Round 1
Reviewer 1 Report
Comments and Suggestions for Authors
The authors describe the inflammation-associated development and progression of penile cancer in tumor immune microenvironment (TIME) of penile cancer. The TME is formed by several immune cell engagement of tumor-associated macrophages, cancer-associated fibroblasts, and tumor-infiltrating lymphocytes (TIL), as the TIME cells express pro-inflammatory cytokines and chemokines to promote penile cancer behavior.
They have focused on NF-κB pathway and secreted phosphoprotein 1 (SPP1) in penile cancer pathogenesis and immune check point PD-1 and PD-L1 for the immune escape of penile cancer from surveillance. PD-L1 expression in the known cancers including penile cancer is a a poor prognostic factor with tumor stage and graded levels, and LN metastasis. A inflammatory protein C-reactive protein (CRP) produced liver and neutrophil-to-lymphocyte ratio (NLR) are also increased as prognostic biomarkers in penile cancer. Penile cancer immunotherapy has been suggested by the authors to downregulate the inflammatory management and penile cancer progression.
Unfortunately, the review direction and aim are too broad as they have link the multiple and separated field in cancer development and progression. I agree with the review on the inflammation and penile cancer development/progression.
The present main stream is not focused on the inflammation/cancer direction.
Thus, I have to mention that they integrate the diverse key fields to solid direction.
For example PD-1/PD-L1 is not directly linked to their maisn stream, but immune therapty, not anti-inflammatory responses.
Author Response
Thank you for an in-depth review of our study. We tried to answer all bothering questions and modified the manuscript according to the reviewer’s suggestions.
Reviewer #1:
Unfortunately, the review direction and aim are too broad as they have link the multiple and separated field in cancer development and progression. I agree with the review on the inflammation and penile cancer development/progression.
The present main stream is not focused on the inflammation/cancer direction.
Thus, I have to mention that they integrate the diverse key fields to solid direction.
For example PD-1/PD-L1 is not directly linked to their main stream, but immune therapy, not anti-inflammatory responses.
We acknowledge your recommendation. The subject of penile cancer and inflammation is indeed extensive and intrinsically linked with immunological factors. To address this matter comprehensively, we have conducted a thorough revision of our manuscript.
First of all we changed the tittle from: Inflammation in Penile Cancer: Current Understanding and Future Directions to Inflammatory Biomarkers and Immunological Targets in Penile Cancer: Current Evidence and Future Directions.
Moreover, this study enhances the section titles by implementing specific, point-by-point headings that effectively engage the reader's attention. Furthermore, the order of the individual sections has been restructured to improve their cohesion and logical flow.
- Tumor-Infiltrating Lymphocytes, Macrophages, and Fibroblasts: Key Players in the Cancer Immune Microenvironment
- The Role of Pro-inflammatory Cytokines and Chemokines in Penile Cancer Progression and Prognosis
- Exploring NF-κB Pathway Activation and Its Implications
- The Secreted Phosphoprotein 1 (SPP1) Gene: From Bone Mineralization to Penile Cancer Prognosis
- C-Reactive Protein as a Biomarker in Cancer: Implications for Penile Cancer Prognosis and Metastasis
- The Neutrophil-to-Lymphocyte Ratio as a Prognostic Biomarker in Penile Cancer
- Expression of PD-L1 in Penile Cancer: Associations with Clinicopathological Features and Patient Outcomes
- Immunotherapy in Advanced Penile Cancer: Current Trials and Clinical Outcomes
To better correlate sections, we provide a brief introduction preceding each section entitled i.e. :
Expression of PD-L1 in Penile Cancer: Associations with Clinicopathological Features and Patient Outcomes
“..” The elevated TILs count is frequently associated with improved clinical outcomes in various malignancies, including penile cancer, as elucidated in the subsection on Neutrophil-to-Lymphocyte Ratio as a Prognostic Biomarker. However, the correlation between TILs count and cancer prognosis is complex, as multiple factors can influence TILs function and efficacy against neoplastic cells. One of the most significant regulatory mechanisms affecting TILs activity is the PD-1/PD-L1 axis. This immune checkpoint pathway can suppress TILs function upon activation, enabling neoplastic cells to evade immune surveillance [166] (Fig.4). “…”
Main text, page 25, lines 877-883
Finally, we present five new figures that elucidate the interconnections among inflammatory biomarkers and immunological targets. (Figure 1-5)
Reviewer 2 Report
Comments and Suggestions for Authors
Czajkowski et al. present a narrative review of a topic of interest and with great repercussions. However, some points need to be improved:
-The authors make a very brief introduction to the state of the art without major approximations so that the specialized reader can focus on the topic. A precise mechanistic approach to specific disease processes with specialized references is necessary. In this sense, biomarkers and cohort studies such as doi: 10.14670/HH-18-846 should be included.
-The authors should improve the titles of the sections, trying to give specific point-by-point messages that attract the reader's attention.
-The authors only include figure 1. Figure 1 is simplistic. It should be improved with references to each of the pathological processes involved.
-The authors should include more figures that represent the pathogenic mechanisms involved in each point. -Section 3 needs to be improved and there should be a precise interconnection. In the current state, it is too simplistic.
-Section 8 is too discrete in the aspects specific to this pathway. It should be more systematic and go point by point.
-Table 1 should be better explained.
-The authors should include all existing clinical trials. Also, those of the Asian population.
-A very important point that should be included in this narrative review is the formation of the inflammasome. This point is important.
Comments on the Quality of English LanguageThe English could be improved to more clearly express the research.
Author Response
The authors express their appreciation for the thorough review of the study. Every effort has been made to address all relevant inquiries and revise the manuscript in accordance with the reviewer's recommendations.
Reviewer #2:
- The authors make a very brief introduction to the state of the art without major approximations so that the specialized reader can focus on the topic. A precise mechanistic approach to specific disease processes with specialized references is necessary. In this sense, biomarkers and cohort studies such as doi: 10.14670/HH-18-846 should be included.
We appreciate your recommendation. The manuscript has been comprehensively revised, incorporating a mechanistic approach to specific disease processes.
“…” Tumor-infiltrating lymphocytes (TILs), a heterogeneous group primarily comprising T, B, and NK cells, play a crucial role in modulating the immune response associated with cancer. The composition of this cell mixture significantly influences the progression or suppression of tumor cells, and thus the prognosis and treatment outcomes in various cancers. The mechanism of action is typically complex, and the relationships among the various types of lymphocytes are multifaceted. Cytotoxic CD8+ T cells play a crucial role in the immune response against tumor cells. These specialized cells recognize specific antigens present on the surface of tumor cells, triggering a complex immune cascade. The process follows a characteristic pathway, involving the initial recognition, activation, and subsequent elimination of the target cells [24]. CD8+ T cells in the TIME are typically supported by CD4+ T helper 1 (Th1) cells, which secrete important cytokines such as interferon-gamma (IFN-γ) and interleukin-2 (IL-2). (Figure 1) These cytokines enhance the activation and proliferation of CD8+ T cells, further amplifying the anti-tumor immune response [19,25]. While CD8+ T cells and Th1 cells are primarily associated with anti-tumor immunity, other CD4+ T cell subsets (Th2) contribute to the overall immune landscape in diverse ways. Th2 cells, for instance, support the B cell response through the production of cytokines such as IL-4, IL-5 which are involved in humoral immunity [24]. In contrast, Th17 cells secrete a distinct set of cytokines, including IL-17, IL-21, and IL-22 which promote tumor growth by creating a favorable environment for cancer cells [26]. This phenomenon illustrates the complex and occasionally contradictory roles that different T cell subsets can exhibit in the context of tumor immunology, emphasizing the need for a nuanced understanding of immune cell interactions within the TIME. The B lymphocytes in the TIME play complex and sometimes contradictory roles in cancer progression. These cells can influence tumor cell survival, proliferation, and treatment resistance, while also potentially facilitating immune escape mechanisms. The precise impact of B cells on cancer development and tumor suppression remains a subject of debate within the scientific community. However, research has demonstrated that modulating B cell activity in the TIME can disrupt cancer-induced immunosuppressive events. One such example is the TGF-β-dependent conversion of FoxP3+ cells, which can support and promote metastasis [27,28]. In contrast, natural killer (NK) cells demonstrate anti-tumor functions. These immune cells are equipped with a variety of receptors that enable them to recognize and eliminate tumor cells while sparing healthy tissues. NK cells serve as a crucial component of the body's initial defense against cancer, utilizing their cytotoxic capabilities to directly eliminate malignant cells mainly by secreting TNF-a, Granzyme A, Granzyme B and perforine [29] (Figure 1). “…”
Main text, page 4-5, lines 146-176
“…” The role of macrophages in TIME is also multifaceted and complex. Tumor-associated macrophages (TAMs) primarily originate from peripheral inflammatory monocytes. (Figure 1) These monocytes are recruited to the tumor site through a complex interplay of the cytokines and chemokines secreted by the tumor cells. Key mediators in this process include colony stimulating factor 1 (CSF-1) and members of the vascular endothelial growth factor (VEGF) family, which function as cytokines, as well as chemokines such as C–C motif chemokine 2 (CCL) 2 and CCL5. (Figure 1) Upon stimulation, these monocytes migrate through the bloodstream and infiltrate the tumor tissues, where they undergo further differentiation to become TAMs. While CSF-1 recruits monocytes to tumors, promotes macrophage survival, and induces TAMs to suppress the immune system, granulocyte–macrophage colony stimulating factor (GM-CSF) exhibits a contrasting effect by stimulating macrophage proliferation and activating anti-tumor functions [40,41] (Figure 1).Previously, it was hypothesized that their primary function was to contribute to antitumor immunity. However, clinical evidence suggests that TAMs can promote tumor growth and malignancy [42]. TAMs are categorized into two populations: M1 and M2.( Figure 1) Interferon-gamma (IFN-γ) plays a crucial role in the polarization of tumor-associated macrophages (TAMs) towards an M1-like phenotype in the TIME. This polarization is characterized by the induction of a Th1-type immune response, which is essential for the innate host defense and antitumor immunity. TAMs M1 exhibit enhanced antigen presentation capabilities through increased expression of major histocompatibility complex class I and II molecules (MHC-I and MHC-II), as well as co-stimulatory molecules CD80 and CD86. These macrophages also produce pro-inflammatory cytokines, including tumor necrosis factor-alpha (TNF-α), and release inflammatory mediators such as reactive oxygen species (ROS) and nitric oxide, contributing to their antitumor functions [19,41,43]. (Figure 1) In contrast, TAMs of the M2 phenotype are polarized in response to cytokines associated with Th2-mediated immune responses, such as IL-4 and IL-10. This polarization significantly impacts various physiological processes, including angiogenesis, anti-inflammatory responses, and immune regulation. M2 macrophages are identified by specific markers like the mannose receptor (CD206) and scavenger receptor, while expressing reduced levels of MHC-II. These cells produce a diverse array of molecules, including arginase-1, matrix metalloproteinase-9 (MMP-9), vascular endothelial growth factor (VEGF), prostaglandin E2 (PGE2), transforming growth factor-beta (TGF-β), and anti-inflammatory cytokines such as IL-10, and IL-13. (Figure 1) The production of these factors contributes to tumor development through multiple mechanisms, highlighting the complex role of TAMs in the tumor microenvironment and their potential as targets for cancer therapy [44,45]. Both types participate in creating an environment that supports cancer invasion and spread. Nevertheless, TAMs M1 generally act as anti-tumorigenic agents, in contrast to TAMs M2, which suppress the immune system and thus encourage cancer development [19]. Elevated TAMs levels are associated with adverse outcomes in various cancers. Allison et al. reported that an elevated concentration of TAMs was associated with diminished OS or PFS in breast cancer. Moreover, their findings indicated that TAMs expressing CD163+ were more likely to predict survival outcomes than those expressing CD68+ [46]. Similarly, Lin et al. observed that elevated TAMs levels were correlated with lymph node metastasis and FIGO stages in cervical cancer. Subsequently, they suggested that a high concentration of TAMs is indicative of poor prognosis in cervical cancer [47]. In squamous cell carcinomas of the head and neck, elevated levels of CD68+CD163+ TAMs are associated with decreased overall survival (OS). Moreover, in these cancers, CD163+ serves as a robust prognostic indicator and is correlated with disease-free survival (DFS) and progression-free survival (PFS) [48].”…”
Main text, page 6-7, lines 211-254
“…” Fibroblasts primarily originate from primitive mesenchyme, with some derived from neural crests [51]. Fibroblasts are major producers of extracellular matrix (ECM) and play crucial roles in tissue repair and wound healing. They can influence epithelial stem cell behavior, promote angiogenesis, and coordinate immune system functions. Fibroblastic reticular cells (FRCs) in lymph nodes create ECM conduits for antigen transit and leukocyte migration. Stellate cells, a specific type of fibroblast in the liver and pancreas, are involved in metabolic homeostasis. Moreover, fibroblasts communicate with various cell types, performing diverse functions beyond ECM production [52]. The TIME contains various mechanisms that trigger the activation of Cancer-Associated Fibroblasts (CAFs). Among these, the most important pathways are initiated by TGF-β, reactive oxygen species (ROS), pro-inflammatory cytokines (such as IL-1 and IL-6), and DNA damage [53] (Figure 1).
The difficulty of defining and characterizing CAFs highlights the challenges faced in understanding their role in tumor progression. While certain criteria have been established to identify CAFs, such as their elongated morphology, the absence of epithelial, endothelial, and leukocyte markers, and a lack of cancer-specific mutations, these parameters may not be sufficient to fully capture the heterogeneity and dynamic nature of these cells. The distinction between CAFs and other mesenchymal cell types, as well as cancer cells that have undergone epithelial-to-mesenchymal transition, further complicates their identification. The primary functions of CAFs include depositing and remodeling the ECM, supporting metastasis by producing supportive matrix components, promoting tumor growth through growth factor and cytokine production, exerting immunosuppressive effects via cytokine production and antigen cross-presentation, and contributing to blood vessel formation through VEGF expression [52,54]. The study coducted by Goulet et al. provides significant insights into the role of cancer-associated fibroblasts (CAFs) and interleukin-6 (IL-6) in bladder cancer progression. Their findings demonstrate that CAFs are a primary source of IL-6 in the TIME, while bladder cancer cells express the IL-6 receptor (IL-6R). This paracrine signaling axis appears to be crucial for promoting tumor growth and metastasis. The researchers observed that exposing bladder cancer cells to CAF-conditioned medium significantly enhanced their proliferation, migration, and invasive capabilities, underscoring the importance of CAF-derived factors in cancer progression. Additionally, the study revealed a correlation between elevated IL-6 expression and more aggressive forms of bladder cancer, as well as an association with increased CAF presence in the tumor microenvironment [55]. Lopez et al. reported that in clear cell renal cell cancer (RCC), fibroblast activation protein (FAP) was associated with increased tumor size, higher grade, and more advanced stage. Moreover, FAP expression was correlated with decreased survival rates [56]. Similarly, Errate et al. demonstrated a significant correlation between FAP immunostaining in the primary tumors of clear cell RCC and several adverse prognostic factors, including advanced tumor stage, high grade, and the presence of necrosis. Notably, FAP expression was also associated with decreased overall survival in clear cell RCC patients. These findings were consistent across various patient subgroups, including those with metastases at diagnosis and those who developed metastases during follow-up, underscoring the potential significance of FAP as a prognostic marker in clear cell RCC [57].
In the literature, there are scarce data regarding the prognostic value of CAF expression in penile cancer. There is only one study on penile cancer and CAF. The study conducted by Cury et al. on 63 penile cancer samples elucidates the critical roles of immune cells and cancer-associated fibroblasts (CAFs) in creating the TIME through the expression and potential secretion of inflammatory factors and extracellular matrix (ECM) remodeling proteinases. The observed negative correlation between immune cell proportions and CAFs in penile cancer samples suggests a complex interplay between these cellular components within the TIME. Furthermore, the analysis demonstrated that patients with elevated CAF scores exhibited reduced survival rates, indicating the prognostic significance of CAFs in penile cancer. This association was accompanied by an increased expression of matrix metalloproteinases and collagens in high-CAF samples. These findings indicate the potential role of CAFs in promoting tumor progression and metastasis through ECM remodeling and the creation of a more permissive environment for cancer cell growth and invasion [58]. “…”
Main text, page 8-9, lines 268-318
“…” The microenvironment of tumors can be classified into two categories: inflamed and non-inflamed. The differences between these two categories are substantial and are primarily influenced by the mixture of immunological elements that they contain. Pro-inflammatory cytokines such as interleukin-1 alpha (IL-1 α), interleukin-1 beta (IL-1β), interleukin-2 (IL-2), interleukin-12 (IL-12), interleukin-23 (IL-23), tumor necrosis factor alpha (TNF-α), and interferon (INFs) types I and II are the primary constituents of the inflammatory microenvironment. This type of immune microenvironment is conducive to T cell activation and expansion. In contrast, the non-inflamed microenvironment is characterized by cytokines that promote immune suppression or tolerance, such as interleukin-4 (IL-4), interleukin-10 (IL-10), interleukin-13 (IL-13), interleukin-17 (IL-17), programmed death-ligand 1 (PD-L1), and transforming growth factor beta (TGF-β) [6,11,63]. (Figure 2) Current research on pro-inflammatory cytokines explains that IL-1A, IL-1B, IL-6, TGF-β1, and IFN-γ might be key components of the penile cancer TIME and are believed to play a significant role in cancer formation [43]. Consequently, these cytokines may contribute to an inflamed penile cancer TIME. Conversely, several studies confirm higher expression of PD-L1 in penile cancer, which contributes to immune suppression and tolerance, thereby potentially creating a non-inflamed TIME [64]. (Figure 2)
Interleukin-1 (IL-1A and IL-1B) and interleukin-1 receptor antagonist (IL-1Ra) are cytokines secreted by macrophages, keratinocytes, and endothelial cells, and they play a role in inflammatory and fibrotic processes.. Additionally, IL-1 plays a significant role in tumor growth and metastasis through its diverse biological activities [65]. The IL-1 family comprises three members: IL-1A and IL-1B, which are active, and interleukin-1 receptor antagonist (IL-1Ra), which functions as an antagonist. Although encoded by separate genes, exhibit identical biological functions, promoting inflammation and tumor progression. In contrast, IL-1Ra binds to the IL-1 receptor without inducing signal transduction, effectively blocking the biological response typically associated with IL-1. The IL-1 signaling pathway involves two receptors from the immunoglobulin superfamily: interleukin 1 receptor type 1 (IL-1R1) and interleukin 1 receptor type 2 (IL-1R2). IL-1R1, which is ubiquitously expressed, is biologically active and responsible for transmitting the IL-1 signal. Conversely, IL-1R2 functions as an IL-1 scavenger, regulating the availability of the cytokine. When IL-1 binds to IL-1R1, it induces the recruitment of the IL-1 receptor accessory protein (IL-1RaP), forming a heterodimeric complex. The formation of this complex initiates the IL-1 signaling cascade, resulting in the activation of various transcription factors. These transcription factors subsequently regulate the expression of genes involved in inflammation, cell proliferation, and survival, thereby contributing to the pro-tumorigenic effects of IL-1 [66,67]. Interleukin-1 plays a significant role in tumor growth and metastasis by promoting metastasis and angiogenesis through the induction of various genes and factors. Specifically, IL-1 upregulates matrix metalloproteinases (MMPs), which facilitate the degradation of extracellular matrix components, allowing cancer cells to invade surrounding tissues and enter the bloodstream. Concurrently, IL-1 enhances the expression of endothelial adhesion molecules, which aid in the attachment of circulating tumor cells to blood vessel walls, a critical step in the metastatic cascade. These combined effects make significant contribution to the dissemination of cancer cells to distant sites within the body. In addition to its pro-metastatic effects, IL-1 is a potent inducer of angiogenesis, the formation of new blood vessels. This process is essential for tumor growth and metastasis, as it provides the necessary oxygen and nutrients to sustain rapidly dividing cancer cells. IL-1 achieves this by stimulating the production of various pro-angiogenic factors, including vascular endothelial growth factor (VEGF), chemokines, and prostaglandin E2 (PGE2). Furthermore, IL-1 promotes the release of growth factors and transforming growth factor-beta (TGF-β), which not only support angiogenesis but also contribute to tumor progression by modulating the tumor microenvironment and suppressing anti-tumor immune responses [67–70]. The combined action of these IL-1-induced factors creates a favorable environment for tumor growth, invasion, and metastasis. According to León et al. the study which enrolled 154 patients with head and neck cancer found that IL-1A overexpression was associated with higher risk of distant metastastasis [71]. The study conducted by Chen et al. found that expression of IL-1B was elevated in esophageal squamous cell carcinoma (SCC) compared to nonmalignant tissues. To elucidate the significance of IL-1B in esophageal squamous cell carcinoma, investigators examined its effects on tumor cell growth, invasion, and treatment response through the regulation of IL-1B signaling. In vitro experiments demonstrated that inhibiting IL-1B signaling with an IL-1B antibody significantly reduced tumor cell growth and invasion ability. Furthermore, the inhibition of IL-1B signaling enhanced cell death induced by cisplatin and irradiation, suggesting that IL-1B may play a crucial role in tumor behavior and treatment resistance. To assess the clinical relevance of these findings, the predictive power of IL-1β on clinical outcomes and treatment response in patients with esophageal SCC was evaluated using immunohistochemistry (IHC) analysis. Moreover, to evaluate the clinical significance of these findings, the predictive capacity of IL-1B regarding clinical outcomes and treatment response in patients with esophageal SCC was assessed utilizing immunohistochemistry (IHC) analysis. Additionally, positive immunohistochemistry (IHC) staining for IL-1B exhibited a significant association with a reduced response to neoadjuvant chemoradiotherapy and inferior clinical outcomes [72]. Similarly, in a study conducted by Elaraj et al., the IL-1 mRNA exhibited elevated expression in metastatic samples and cancer cell lines derived from patients with non-small-cell lung carcinoma, colorectal adenocarcinoma, and melanoma [73].
Interleukin-6 (IL-6) is a multifunctional cytokine that plays a crucial role in the immune system and various physiological processes. Initially identified as a B-cell stimulatory factor, IL-6 has since been recognized for its diverse range of biological activities. This 26-kD secreted protein not only promotes antibody production in B cells but also influences T-cell differentiation, hematopoiesis, and acute-phase protein synthesis in the liver. The pleiotropic nature of IL-6 is further exemplified by its involvement in inflammation, metabolism, carcinogenesis, and neural processes [62]. The IL-6 signaling pathway is complex and involves multiple components, including the IL-6 receptor (IL-6R) and the signal transducer glycoprotein 130 (gp130). IL-6 can activate cells through two distinct mechanisms: classical signaling and trans-signaling. In classical signaling, IL-6 binds to membrane-bound IL-6R, which then associates with gp130 to initiate intracellular signaling cascades. Trans-signaling, conversely, involves a soluble form of IL-6R that can bind IL-6 and activate cells that do not express membrane-bound IL-6R. This dual signaling capability enables IL-6 to exert its effects on a wide range of cell types and contributes to its diverse biological functions. The overlapping activities of IL-6 with other members of the IL-6 family of cytokines, which also utilize gp130 for signal transduction, further underscore the complexity and significance of this signaling network in maintaining homeostasis and mediating various physiological responses [62,74].The research by Gilabert et al. revealed a significant correlation between elevated levels of IL-6 expression and poor prognosis in pancreatic cancer patients. The worsened prognosis was attributed to the promotion of cancer development accompanied by cachexia [75].The elevated levels of IL-6 in the serum have been identified as a significant factor in the progression and prognosis of colorectal cancer. This pro-inflammatory cytokine plays a crucial role in promoting epithelial to mesenchymal transition (EMT) which provide to loss of cell-cell adhesion and increased motility, allowing cancer cells to detach from the primary tumor site and invade surrounding tissues. As a result, elevated IL-6 levels contribute to enhanced tumor invasiveness and metastatic potential in colorectal cancer patients. Furthermore, increased serum IL-6 levels have been correlated with larger tumor sizes and a higher likelihood of metastasis in colorectal cancer. Consequently, these factors contribute to decreased survival rates among colorectal cancerpatients [76].Similarly, a study of 85 patients with muscle-invasive bladder cancer revealed that IL-6 was expressed at higher levels in bladder cancer compared to non-malignant tissues. Furthermore, positive staining for IL-6 was primarily associated with muscle-invasive bladder cancer relative to lower stage Ta–T1 disease. Additionally, urinary levels of IL-6 were also significantly elevated in patients with locally advanced bladder TCC compared to patients with NMIBC. Consequently, IL-6 expression may be associated with a more malignant phenotype [77].
The transforming growth factor-β (TGF-β) family plays a crucial role in regulating diverse cellular processes and has significant implications in organism homeostasis and disease. This family of proteins, which includes TGF-β, activins/inhibins, and bone morphogenic proteins (BMPs), exerts its effects through a complex signaling cascade. When TGF-β ligands bind to their receptors, they initiate a series of phosphorylation events, ultimately leading to the activation of SMAD proteins. These activated SMADs translocate to the nucleus, where they regulate the transcription of target genes, influencing various cellular functions such as proliferation, apoptosis, differentiation, EMT and migration [78–80]. The role of TGF-β in cancer development is particularly noteworthy due to its context-dependent effects. In the early stages of carcinogenesis, TGF-β functions as a tumor suppressor by inhibiting cell cycle progression and promoting apoptosis. However, as cancer progresses, TGF-β undergoes a functional transition, becoming a promoter of tumor growth, invasiveness, and metastasis. This dual nature of TGF-β signaling underscores the complexity of its role in cancer biology. Furthermore, the TGF-β pathway interacts with other signaling pathways in both synergistic and antagonistic manners, contributing an additional layer of complexity to its regulatory functions. The clinical significance of TGF-β activity is emphasized by the association between elevated levels of TGF-β and poor prognosis in various cancers, rendering it an important target for therapeutic interventions and biomarker development in oncology [81,82].The study by Hegele et al. compared plasma levels of latent and active TGF-β1 in patients with localized RCC (n=39), metastatic RCC (n=17), and benign diseases as a control group (n=93). No significant differences in TGF-β1 levels were observed between localized RCC patients and the control group. Similarly, no significant variations in TGF-β1 levels among localized RCC subgroups based on cancer stage and grade were identified. However, significantly higher levels of both latent and active TGF-β1 were found in metastatic RCC patients compared to those with localized RCC. These findings suggest a potential association between elevated TGF-β1 levels and advanced stages of RCC [83].Research conducted by Chen et al. revealed a notable link between the Int7G24A (rs334354) intronic variation in the TGFBR1 gene and an elevated susceptibility to both bladder transitional cell carcinoma (TCC) and RCC. Individuals with heterozygous or homozygous single nucleotide polymorphisms (SNPs) at this locus showed a higher susceptibility to these urological cancers. Additionally, a somatic mutation causing a serine to phenylalanine substitution at codon 57 of TGFBR1 was discovered. These findings suggest a common genetic factor in bladder and kidney cancer predisposition and highlight the potential role of TGF-β signaling pathway alterations in cancer pathogenesis [84].Fang et al. reveals miR-34b as a crucial regulator of the TGF-β pathway in prostate cancer cells. The miR-34b modulates the expression of TGF-β, TGFBR1, pSMAD4, and p53. Upregulation of miR-34b in PC3 prostate cancer cells inhibits cell growth, migration, and invasion. These findings suggest that targeting miR-34b could be a promising therapeutic approach for prostate cancer treatment due to its ability to affect multiple aspects of cancer progression through TGF-β signaling modulation [85].
Interferon-γ (IFN-γ) is a cytokine with diverse biological functions, particularly in immune regulation and host defense against pathogens. As a member of the interferon family, it was initially identified for its capacity to interfere with viral replication. IFN-γ is unique among interferons, being the sole representative of type II interferons. It is produced by various immune cells, including both innate and adaptive immune components, in response to potentially harmful stimuli. This cytokine plays a vital role in numerous physiological processes, extending beyond its primary functions in immune modulation and antimicrobial defense. The impact of IFN-γ extends to various aspects of human health and disease. It is involved in pregnancy, obesity, allergic reactions, and autoimmune disorders [86–88]. IFN-γ has been the subject of extensive research in cancer biology, revealing its complex and occasionally contradictory roles. Initially, it was recognized as a potent anti-tumor agent. However, subsequent studies have uncovered a more nuanced perspective on the influence of IFN-γ on cancer development and progression. While IFN-γ can inhibit tumor initiation and growth, it also plays a role in shaping tumor immunogenicity. Paradoxically, it can promote the emergence of tumor cells with enhanced capabilities to evade immune detection and elimination [88,89]. The comprehensive bioinformatics study utilizing the Cancer Genome Atlas Program (TCGA) database has identified CDKN3 as a potential prognostic indicator in clear cell RCC. Elevated CDKN3 levels correlate with reduced overall survival and unfavorable outcomes in clear cell RCC patients. This effect is partly mediated through the activation of inflammatory pathways, including IL-6/JAK/STAT3, TNF-α/NF-kB, and IFN-γ signaling [90]. In contrast, Otessen et al. investigated the effects of recombinant human interferon γ (rHu-INFγ) on malignant and pre-malignant urothelial cell lines. Their findings revealed that malignant cells experienced significant growth inhibition (>50%) when exposed to rHu-INFγ, while pre-malignant cells showed less sensitivity. The study suggests that rHu-INFγ has potential as a treatment for human bladder cancer, particularly for malignant urothelial cells [91].
“..”
Main text, page 10-14, lines 334-496
“…” The investigation conducted by Pennatochiotti et al. examined 79 oral SCC patients, categorizing cases by severity. The findings demonstrated elevated NFKB1 mRNA and protein levels in more advanced stages of oral SCC, suggesting NFKB1's potential role in cancer progression [120]. Similarly, in a study of 28 patients, Fonseca et al. identified an association between elevated NFKB1 mRNA levels and the severity of oral SCC [121]. The in vitro experiment conducted by Lehman et al. provided valuable insights into the relationship between the canonical NF-κB pathway and metastatic potential in human esophageal epithelial cells. Upon the activation of this pathway, the researchers observed a significant enhancement in the cells' metastatic capacity. This finding suggests that the NF-κB signaling cascade plays role in promoting the aggressive behavior of esophageal cancer cells [122]. “…”
Main text, page 20 lines 686-695
“…” An elevated TILs count is frequently associated with improved clinical outcomes in various malignancies, including penile cancer, as elucidated in the subsection on The Neutrophil-to-Lymphocyte Ratio as a Prognostic Biomarker. However, the correlation between the TIL count and cancer prognosis is complex, as multiple factors can influence TILs function and efficacy against neoplastic cells. One of the most significant regulatory mechanisms affecting TILs activity is the PD-1/PD-L1 axis. This immune checkpoint pathway can suppress TIL function upon activation, enabling neoplastic cells to evade immune surveillance [171] (Figure 4).
Programmed cell death protein 1 (PD-1) is a cell surface receptor expressed on various immune cells, including CD8+ T cells, CD4+ T cells, natural killer (NK) cells, monocytes, antigen-presenting cells (APCs) and CD20+ lymphocytes. PD-1 plays a crucial role in regulating T lymphocyte activity by interacting with its cognate ligand, programmed cell death ligand 1 (PD-L1) [172,173]. PD-L1, a membrane-bound protein, forms a complex with PD-1 on CD8+ T cells, leading to inhibition of immune responses. Remarkably, PD-L1 has been detected in tissues, such as the placenta and pancreatic islets, where it is thought to play a role in maintaining immune tolerance. Under normal circumstances, PD-L1 expression is induced in response to inflammation and elevated cytokine levels to prevent excessive immune reactions [174]. Cancer cells utilize PD-L1 as an immune evasion mechanism to inhibit the host antitumor immune response. Overexpression of PD-L1 on cancer cells leads to the activation of tyrosine phosphatase SHP2 membrane protein in CD8+ cells. Subsequently, due to the inactivation of PI3K/Akt and MEK/EKR pathways, as well as the inhibition of MHC-TCR and CD80-CD28 interactions, it results in CD8+ exhaustion. This process is characterized by the overexpression of PD-1 and elevated expression of CTL4-A, TIM-3, and LAG-G surface proteins, while cytokine secretion levels are decreased. Moreover, CD8+ T cells may undergo apoptosis, or their proliferation and activation rates may decrease [171]. (Figure 4) Therefore the overexpression of PD-L1 in tumor cells can prevent immune recognition and clearance, primarily through the action of CD8+ T cells [175]. Consequently, PD-L1 has been established as a prognostic and predictive factor for multiple types of cancers. Research conducted by Ueda et al. demonstrated that the expression of PD-1 and PD-L1 in metastatic clear cell RCC exhibited positive expression in 48.5% and 27.3% of cases, respectively. This expression correlated with increased infiltration of CD4+, CD8+, and FOXP3+ TILs. Moreover, the positive PD-1 and PD-L1 expression was significantly associated with adverse clinicopathological features, including increased renal tumor size, higher Fuhrman Grade, and the presence of sarcomatoid features. Additionally, PD-1 expression was linked to poor outcomes in metastatic clear cell RCC patients undergoing molecular targeted therapies [176]. Similarly, PD-L1 is expressed in non-small cell lung cancer samples, with its presence detected in both the cancer cell membrane and cytoplasm. A statistically significant correlation was observed between the proportion of PD-L1-positive cells and the cancer's histological type, with adenocarcinoma exhibiting higher levels of expression compared to squamous cell carcinoma. Notably, a greater percentage of PD-L1-expressing lung cancer cells was significantly associated with a poorer prognosis [177]. The investigation conducted by Hino et al. provides significant insights into the prognostic implications of PD-L1 expression in melanoma patients. Through the examination of specimens from 59 individuals with melanoma, the researchers established a correlation between elevated PD-L1 expression on tumor cells and unfavorable patient outcomes. This finding emphasizes the significance of PD-L1 as a biomarker in melanoma and its potential utility in informing treatment decisions [178]. In summary, in the majority of cancers, elevated expression of PD-L1 has been associated with unfavorable pathological features and poor clinical outcomes. “…”
Main text, page 26-27 lines 882-925
Moreover, as reviewer mentioned we added the citation: doi: 10.14670/HH-18-846
“…” The study by Casanova-Martín et al. provided significant insights into the expression of NLRP3 in penile cancer specimens. Their findings demonstrated a correlation between NLRP3 expression levels and tumor differentiation. Specifically, poorly differentiated penile cancer tumors exhibited elevated NLRP3 expression compared to moderately and well-differentiated tumors or verrucous carcinoma. This observation suggests that NLRP3 may play a crutial role in the progression of penile cancer, potentially serving as a marker for more aggressive forms of the disease. This association may also indicate that NLRP3 contributes to the development of more aggressive phenotypes in penile cancer [98] “…”
Main text, page 16, lines 554-561
“…” The study conducted by Casanova-Martín et al. provides significant insights into the role of allograft inflammatory factor 1 (AIF-1) in penile cancer. Their findings reveal that AIF-1 is expressed in penile cancer tissue samples, suggesting its potential involvement in the disease process. AIF-1 participates in carcinogenesis through its ability to activate key signaling pathways, specifically the NF-κB pathway and β-catenin. Furthermore, the researchers observed a correlation between elevated levels of AIF-1 and more aggressive penile cancer tumors. This association suggests that AIF-1 may serve as a potential biomarker for disease severity and progression [98]. “…”
Main text, page 20, lines 707-714
- The authors should improve the titles of the sections, trying to give specific point-by-point messages that attract the reader's attention.
We acknowledge your recommendation. The section titles have been modified through the implementation of specific, point-by-point headings that effectively capture the reader's attention. Additionally, the sequence of individual sections has been reorganized to enhance their coherence and logical progression.
- Tumor-Infiltrating Lymphocytes, Macrophages, and Fibroblasts: Key Players in the Cancer Immune Microenvironment
- The Role of Pro-inflammatory Cytokines and Chemokines in Penile Cancer Progression and Prognosis
- Exploring NF-κB Pathway Activation and Its Implications
- The Secreted Phosphoprotein 1 (SPP1) Gene: From Bone Mineralization to Penile Cancer Prognosis
- C-Reactive Protein as a Biomarker in Cancer: Implications for Penile Cancer Prognosis and Metastasis
- The Neutrophil-to-Lymphocyte Ratio as a Prognostic Biomarker in Penile Cancer
- Expression of PD-L1 in Penile Cancer: Associations with Clinicopathological Features and Patient Outcomes
- Immunotherapy in Advanced Penile Cancer: Current Trials and Clinical Outcomes
- The authors only include figure 1. Figure 1 is simplistic. It should be improved with references to each of the pathological processes involved.
We appreciate your suggestion. We have improved Figure 1 to better illustrate the described processes.
(Figure 1)
- The authors should include more figures that represent the pathogenic mechanisms involved in each point. -Section 3 needs to be improved and there should be a precise interconnection. In the current state, it is too simplistic.
Thank you for your suggestion. We have created 4 new figures to better describe the pathogenic mechanisms. Moreover, 3rd section have been substantially improved. We have incorporated new studies and added a paragraph about inflammasomes.
(Figure 2-5)
The Role of Pro-inflammatory Cytokines and Chemokines in Penile Cancer Progression and Prognosis
Main text, page 9-17, lines 324-614
- Section 8 is too discrete in the aspects specific to this pathway. It should be more systematic and go point by point.
- Thank you for your suggestion. The 8th section have been substantially improved. We tried to make it point by point. Please note that after the reordering, section 8 of the previous version of the manuscript is now numbered 7 (The Neutrophil-to-Lymphocyte Ratio as a Prognostic Biomarker in Penile Cancer)
Main text, page 22-28, lines 770-962
- Table 1 should be better explained
Thank you for your input. We have included an explanation for Table 1 and condensed the main findings. Table 1 offers a comprehensive overview of all studies discussed in the main text that focus on inflammatory molecules and immunological targets in penile cancer.
Table 1, page 32
- The authors should include all existing clinical trials. Also, those of the Asian population.
We appreciate your recommendation. We have created a new table (Table 3) that presents information on all current clinical trials evaluating the effectiveness of immunotherapy for penile cancer. Additionally, we have expanded Table 2 to encompass all original research articles published on immunotherapy in penile cancer.
Table 2 - Immunotherapy in penile cancer. (page 36-37)
Table 3 - Clinical trials investigating efficiency of immunotherapy in penile cancer (page 38-42)
- A very important point that should be included in this narrative review is the formation of the inflammasome. This point is important.
We appreciate your suggestion. We have incorporated the information regarding the inflammasome.
“…” Inflammasomes are complex protein structures that serve as critical components of the innate immune system, acting as intracellular sensors and initiators of inflammatory responses mediated among others proinflammatory cytokines including IL-1β, and IL-18 [94]. These structures are composed of multiple proteins, including a sensor protein (such as NLR family pyrin domain containing 3 - NLRP3, absent in melanoma 2 - AIM2, or NLR family CARD domain-containng protein 4 - NLRC4, an adaptor protein (apoptosis-associated spect-like protein coteining a CARD - ASC), and an effector protein (pro-caspase-1). When activated by various stimuli, including pathogen-associated molecular patterns (PAMPs), damage-associated molecular patterns (DAMPs), or cellular stress signals, these components assemble to form the active inflammasome complex [95]. This process initiates a series of reactions that ultimately result in caspase-1 activation, a crucial enzyme in inflammation. The inflammasome-mediated activation of caspase-1 has widespread implications for immune responses and cellular functions. Caspase-1 cleaves pro-inflammatory cytokines, particularly IL-1β and interleukin-18 (IL-18), into their active forms, which are then released from the cell to propagate inflammation and recruit additional immune cells to the site of infection or injury. Furthermore, caspase-1 catalyzes the cleavage of gasdermin D (GSDMD), a protein that, upon activation, forms pores in the cell membrane. These pores not only facilitate the release of mature cytokines but also result in a rapid form of cell death termed pyroptosis [94,96]. Pyroptosis is characterized by cellular swelling, membrane rupture, and the release of cellular contents, including additional DAMPs, which further amplify the inflammatory response [97]. The ability of inflammasomes to initiate both cytokine release and pyroptosis higlighted their central role in regulating innate immune responses and highlights their importance in various physiological and pathological processes, including infection control, autoimmune diseases, and inflammatory disorders [94,97].
In the literature, only two studies have investigated the overexpression of inflammasome components in penile cancer, including NLRP3 [98] and AIM2 [99].
The NLRP3 inflammasome exhibits a complex role in cancer, with conflicting evidence. Generally, NLRP3 overexpression is associated with poorer outcomes [100]. Research has demonstrated that the NLRP3 inflammasome promotes breast cancer progression and metastasis by triggering the release of IL-1β [101]. Furthermore, the NLRP3 inflammasome plays a crucial role in regulating inflammation in prostate cancer. It contributes to the growth, survival, migration, and invasion of tumor cells by influencing autophagy, mitochondrial metabolism, and EMT [102]. However, some studies have emphasized the protective role of NLRP3 in tumorigenesis. According to Wei et al. NLRP3 expression is upregulated in inflammatory liver tissue but downregulated in cancerous tissue, suggesting that NLRP3 deficiency may contribute to hepatocellular carcinoma progression [103]. The study by Casanova-Martín et al. provided significant insights into the expression of NLRP3 in penile cancer specimens. Their findings demonstrated a correlation between NLRP3 expression levels and tumor differentiation. Specifically, poorly differentiated penile cancer tumors exhibited elevated NLRP3 expression compared to moderately and well-differentiated tumors or verrucous carcinoma. This observation suggests that NLRP3 may play a crutial role in the progression of penile cancer, potentially serving as a marker for more aggressive forms of the disease. This association may also indicate that NLRP3 contributes to the development of more aggressive phenotypes in penile cancer [98].
The AIM2 inflammosome simirarly to NLRP3 demonstrates influence to cancerogenesis, although conflictic data exist. Some studies provide information about AIM2’s involvement in cancerogenesis. Farschian et al. reported that AIM2 expression is notably elevated in both primary and metastatic cutaneous SCC cell lines when compared to normal keratinocytes [104]. In contrast, a study by Dihlmann et al. involving 476 colon cancer specimens demonstrated that the absence of AIM2 expression was associated with decreased survival rates, quicker disease recurrence, and metastatic progression. Furthermore, patients whose tumor cells exhibited complete absence of AIM2 expression faced a mortality risk from disease progression that was more than threefold higher than those whose tumor cells expressed AIM2 [105]. The study conducted by Tan et al. identified 22 upregulated genes in penile cancer tissues, with B cell lymphoma 2-related protein A1 (BCL2A1) and AIM2 primarily associated with cellular proliferation. These findings were corroborated by elevated mRNA levels of BCL2A1 and AIM2 in tumor samples compared to normal tissue. The investigators determined that BCL2A1 and AIM2 are reliable oncogenes in PeCa, with their overexpression correlated with cancer-specific survival (CSS). The combined expression levels of both genes had a significant influence on CSS, with the double-negative group exhibiting the highest five-year CSS rate (80.5%), followed by the single-positive group (68.6%), and the double-positive group (38.9%). Notably, the presence or absence of AIM2 did not affect the secretion of cleaved IL-1β and IL-18 in penile cancer cells, suggesting that AIM2 may not depend on anti-tumor inflammatory cytokines to influence immune response in PeCa cells [99]. „…”
Main text, page 15-17, lines 522-580
- The English could be improved to more clearly express the research.
We revised our manuscript by English language standards. Furthermore, a clean version of the manuscript underwent editing by MDPI English Editing.
Round 2
Reviewer 1 Report
Comments and Suggestions for Authors
The present review deals with penile cancer-associated inflammation. As described, inflammation responses are related with the development and progression of cancers including penile cancer. In addition, the tumor TME is also associated with innate immune tumor-associated macrophages, fibroblasts, and TILs. As the TME generates pro-inflammatory molecules in cancer region. Penile cancer falls into the above conceptual biology with tumor progression. Surely, penile cancer immunotherapy is clearly limited. However, the present review does not cover the needs and requirements to understand the penile cancer biology in this present form. No penile cancer-specific progression in review description is described. What is the distinct phenotype of the present penile cancer. The present review is very limited to justify its publication at this occasion.
Penile cancer-associated or involved tumor biology should be described, but not for general cancer immune therapy by PD/PDL-1, or general known issues.
Reviewer 2 Report
Comments and Suggestions for Authors
The authors present the revised version of the manuscript ijms-3389944. The authors have successfully made all the required changes. The manuscript is now of a high quality. In my opinion, it can be accepted in its current form for publication.